# Intratumoral modulation of the inducible co-stimulator ICOS by recombinant oncolytic virus promotes systemic anti-tumour immunity

Dmitriy Zamarin[1,2,3,4], Rikke B. Holmgaard[3,4], Jacob Ricca[3,4], Tamar Plitt[3,4], Peter Palese[5], Padmanee Sharma[6], Taha Merghoub[2,3,4], Jedd D. Wolchok[1,2,3,4,*] & James P. Allison[6,*]

Emerging data suggest that locoregional cancer therapeutic approaches with oncolytic viruses can lead to systemic anti-tumour immunity, although the appropriate targets for intratumoral immunomodulation using this strategy are not known. Here we find that intratumoral therapy with Newcastle disease virus (NDV), in addition to the activation of innate immunity, upregulates the expression of T-cell co-stimulatory receptors, with the inducible co-stimulator (ICOS) being most notable. To explore ICOS as a direct target in the tumour, we engineered a recombinant NDV-expressing ICOS ligand (NDV-ICOSL). In the bilateral flank tumour models, intratumoral administration of NDV-ICOSL results in enhanced infiltration with activated T cells in both virus-injected and distant tumours, and leads to effective rejection of both tumours when used in combination with systemic CTLA-4 blockade. These findings highlight that intratumoral immunomodulation with an oncolytic virus expressing a rationally selected ligand can be an effective strategy to drive systemic efficacy of immune checkpoint blockade.

[1] Department of Medicine, Memorial Sloan Kettering Cancer Center, New York, New York 10065, USA. [2] Weill Cornell Medical College, New York, New York 10065, USA. [3] Ludwig Collaborative Laboratory, Memorial Sloan Kettering Cancer Center, New York, New York 10065, USA. [4] Swim Across America Laboratory, Memorial Sloan Kettering Cancer Center, New York, New York 10065, USA. [5] Department of Microbiology, Icahn School of Medicine at Mount Sinai, New York, New York 10029, USA. [6] Department of Immunology, MD Anderson Cancer Center, Houston, Texas 77030, USA. * These authors contributed equally to this work. Correspondence and requests for materials should be addressed to D.Z. (email: Zamarind@mskcc.org).

The immune system plays a key role in eliminating and containing early tumour growth. Tumour progression occurs as a result of cancer cells acquiring the ability to escape immune surveillance through a variety of mechanisms[1–3]. These include downregulation of tumour-associated antigens, enhanced resistance to apoptotic stimuli and alteration of the local tumour microenvironment. In addition, tumours may utilize additional immunosuppressive pathways, which normally act to limit T-cell responses. These include upregulation of the inhibitory CTLA-4 and PD-1 receptors on lymphocytes, as well as direct tumour expression of inhibitory ligands such as PD-ligand 1 (PD-L1), B7-H3 and B7x (ref. 4).

Targeting of inhibitory immune checkpoints for cancer therapy has demonstrated durable responses, though clinical benefit has been limited to subsets of patients within a few, but growing number, of cancer types. Such challenges in immunotherapy logically call for the development of combinatorial approaches and analysis of markers predicting better response[5,6]. Indeed, in melanoma, recent studies indicate significant enhancement of activity of PD-1 blockade when combined with CTLA-4 blockade, an effect that was primarily seen in patients with PD-L1-non-expressing tumours[7]. The reported activity, however, was still not universal and significant toxicities reported from the CTLA-4/PD-1 combination regimen create challenges in building further treatment combinations based on this platform, logically calling for the identification of additional targetable markers and development of rational combinatorial approaches that would minimize toxicity.

Locoregional therapeutic approaches may enhance the efficacy of systemic immune checkpoint blockade, while potentially avoiding additional systemic toxicity. Clinical studies are currently ongoing combining radiation to a focal lesion with systemic CTLA-4 blockade in an attempt to explore the frequency of induction of so-called abscopal responses[8]. Several studies have also demonstrated that intratumoral administration of TLR agonists could be effective against distant tumours[9]. These findings highlight that targeting of immune pathways through combinations of both locoregional and systemic immunotherapeutic approaches may be required for optimal therapeutic efficacy. To this end, oncolytic viruses (OVs) present an attractive strategy for locoregional immune activation, leading to immunogenic cell death, antigen release and production of type I interferon (IFN)—all factors required for efficient dendritic cell (DC) maturation and cross-presentation of tumour antigens[10–12]. We have explored this strategy using oncolytic Newcastle disease virus (NDV) and demonstrated that localized tumour infection with NDV-induced lymphocytic infiltration within virus-injected and distant tumours, resulting in regression of all tumours when combined with systemic CTLA-4 blockade[13]. This effect is not limited to NDV and recent studies have demonstrated that other OVs could be similarly used to potentiate the efficacy of immune checkpoint blockade[14–16].

In addition to the enhancement of antigen release and presentation, intratumoral approaches with OVs provide an opportunity to target-specific immune pathways directly within tumours, thus potentially avoiding systemic toxicity. To date, the optimal pathways for direct intratumoral targeting are not known and may involve components of both the innate and adaptive immune systems. Furthermore, the choice of a target may be further influenced by the other immunotherapeutic agents administered concurrently. With OV therapy, such targets are governed by complex interactions of a specific OV with the tumour microenvironment, and are influenced by individual virus biology, its replication and lytic potential, and its effects on the tumour cells and stromal cells.

In the current study, we set out to characterize relevant pathways activated in response to intratumoral NDV therapy and to determine whether such pathways could be targeted directly within the tumour microenvironment using a recombinant ligand expressed by the virus. We hypothesized that enhancement of T-cell effector function within the tumour microenvironment through a relevant co-stimulatory pathway may drive a better anti-tumour immune response. To this end, here we have identified the inducible co-stimulator (ICOS) as a pathway upregulated in NDV-infected tumours and investigated it as a target using recombinant NDV expressing the ICOS ligand (ICOSL) directly within the tumour microenvironment. We demonstrate that this strategy can significantly augment the efficacy of immune checkpoint blockade, providing a strong rationale for its evaluation in clinic.

## Results

**NDV upregulates ICOS in tumour microenvironment.** To characterize the local and abscopal effects of intratumoral NDV therapy, we used the bilateral flank B16-F10 melanoma model, with the virus administered to a unilateral tumour (Fig. 1a). Consistent with our previous findings, localized administration of a naturally occurring attenuated NDV led to an inflammatory response, resulting in the local and distant tumour infiltration with CD8[+] and CD4[+]FoxP3[−] conventional T cells (Tcon), with significant reduction in the relative percentage of CD4[+]FoxP3[+] regulatory T cells (Tregs; Fig. 1b–e). Analysis of the treated tumour-draining lymph nodes revealed expansion of T-cell populations and increased infiltration with myeloid DCs and CD8[+] DCs, known for their critical role for antigen cross-presentation in tumours[17,18] (Fig. 1f). Consistent with an increased inflammatory response in tumours, combining intratumoral NDV with systemic CTLA-4 blockade led to regression of both virus-injected and distant tumours[13]. Therapeutic efficacy of this approach, however, became attenuated when a larger tumour challenge was used (Supplementary Fig. 1a,b), with <40% of the animals having long-term regression of distant tumours. We hypothesized that targeting of additional immune pathways within the NDV-treated tumours could enhance anti-tumour immunity and drive therapeutic efficacy within the context of larger tumours. To define the immune pathways upregulated by NDV, RNA isolated from NDV-treated tumours was analysed on a NanoString platform using the nCounter PanCancer Immune Profiling Panel. Infection with NDV resulted in upregulation of the majority of the immune-related genes in the panel (Fig. 1g), with induction of genes related to type I IFN, DC function and T-cell function (Supplementary Fig. 2; Supplementary Tables 1–3). The immune transcriptional signatures defining individual cell subsets were consistent with the upregulation of cytotoxic and Th1, but not Th2 responses (Supplementary Fig. 3).

Gene expression profiling thus identified a range of immune pathways that could be targeted within the context of NDV therapy. Strategies using engineered oncolytic DNA viruses such as vaccinia and herpesvirus to date have primarily focused on enhancing the recruitment of antigen-presenting cells (APCs) through production of granulocyte–macrophage colony-stimulating factor (GM-CSF)[19,20]. Given the existing strong recruitment and activation of DCs and type I IFN with wild-type NDV (Fig. 1f; Supplementary Fig. 2), we chose to explore targets that could potentially more directly enhance T-cell activation in this study. We specifically focused on known T-cell co-stimulatory receptors that have been previously demonstrated to be viable targets for systemic agonist immunotherapy. Assessment of expression of genes encoding the TNF receptor superfamily

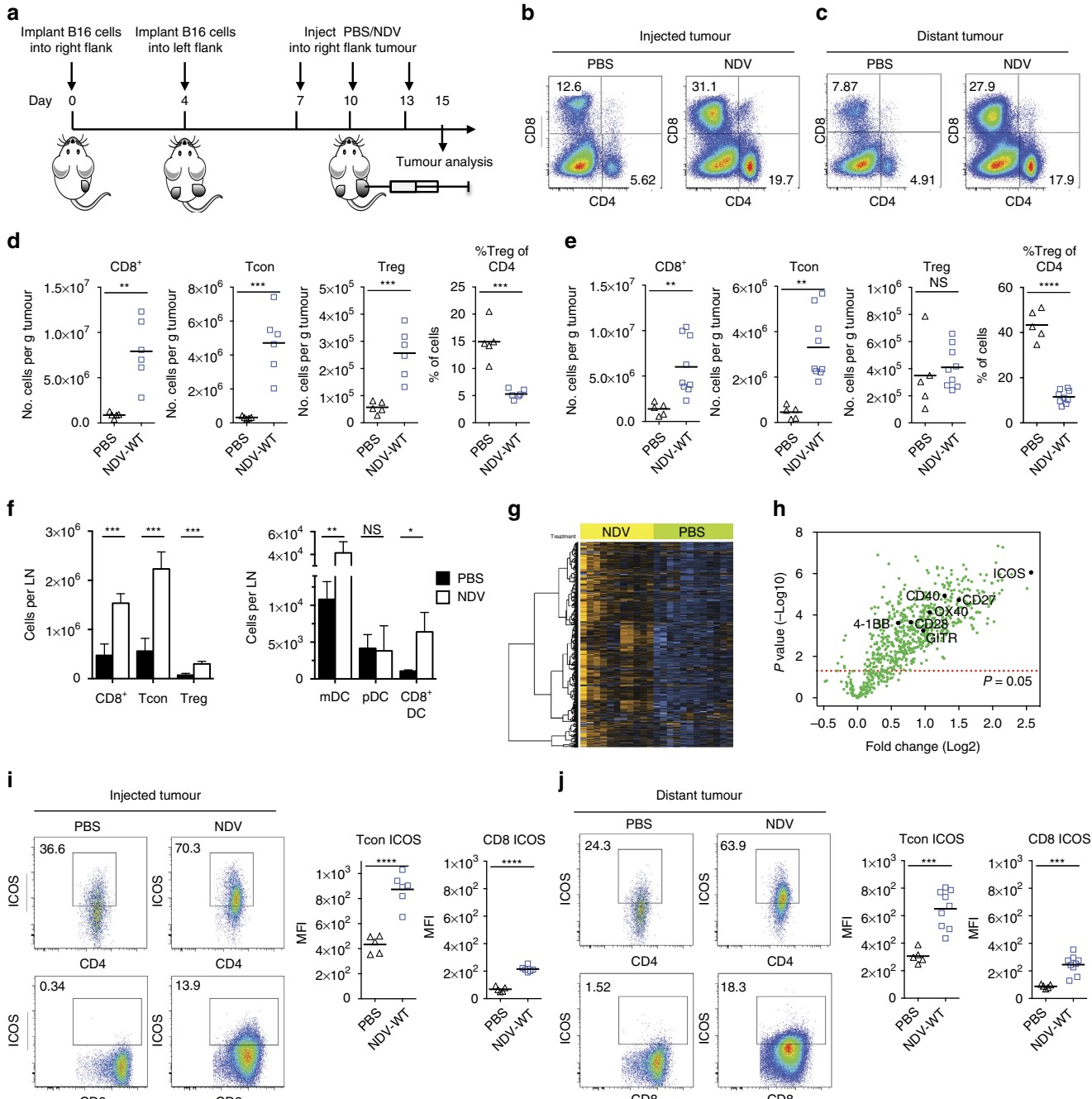

**Figure 1 | NDV induces upregulation of ICOS in the tumour microenvironment.** Bilateral flank B16-F10 melanoma-bearing mice were treated intratumorally with NDV. (**a**) Treatment scheme. (**b,c**) Representative flow cytometry plots of tumour-infiltrating CD4+ and CD8+ cells isolated from the virus-injected (right) and distant (left) tumours, gated on the total CD45+ cell population. (**d,e**) Absolute number of tumour-infiltrating CD8+, CD4+FoxP3− (Tcon) and CD4+FoxP3+ (Treg) lymphocytes isolated from the virus-injected (**d**) and distant (**e**) tumours. Relative percentages of Tregs are shown in the rightmost panels. (**f**) Absolute numbers of lymphocytes (left) and dendritic cell subsets (right) in the lymph nodes draining virus-injected tumours. (**g**) Expression of immune-related genes in NDV-injected tumours. Orange indicates high expression and blue indicates low expression. (**h**) Volcano plot of gene expression in NDV-injected tumours. Genes related to T-cell co-stimulatory function are indicated. (**i,j**) Upregulation of ICOS on the surface of tumour-infiltrating CD8 and CD4+FoxP3− cells in the NDV-injected (**i**) and distant (**j**) tumours. *P<0.05, **P<0.01, ***P<0.001, ****P<0.0001, NS, non-significant. mDC, myeloid dendritic cell; pDC, plasmacytoid dendritic cell; Tcon, conventional CD4 T cell (CD4+FoxP3−); Treg, regulatory T cell (CD4+FoxP3+). (**b–j**) Representative data of three experiments with 5–10 animals per group. Data with error bars represent mean ± s.e.m. Two-tailed Student's *t* test was used in all comparisons.

members as well as the immunoglobulin receptor superfamily revealed upregulation of genes encoding CD28, ICOS, 4-1BB, GITR, OX40, CD27 and CD40 within the NDV-injected tumours, with ICOS being most prominent (Fig. 1h). Analysis

of lymphocytes isolated from both NDV-injected and distant tumours confirmed the upregulation of ICOS on both CD4+ FoxP3− and CD8+ cells (Fig. 1i,j). These findings created a rationale for targeting of ICOS directly in the tumour, which was

also supported by our prior data demonstrating that ICOSL-expressing cellular vaccine could potentiate the efficacy of CTLA-4 blockade[21].

**Generation and *in vitro* evaluation of NDV-ICOSL virus**. To target ICOS directly within tumours, we engineered a recombinant NDV-expressing full-length murine ICOSL (NDV-ICOSL), with the transgene inserted between the viral P and M genes (Fig. 2a)[22]. We proceeded to select an appropriate model for evaluation of the recombinant virus. In our previous studies, we have found that intratumoral NDV could potentiate the efficacy of systemic CTLA-4 blockade in the B16 model, which reasonably supports NDV infection, as well as TRAMP C2 prostate adenocarcinoma and CT26 colon carcinoma, which have poor permissiveness for NDV[13]. While these previous findings suggested that robust NDV replication and lysis were not necessary for generation of anti-tumour immunity, for the purposes of the current study, we screened for a model that could support good levels of therapeutic transgene expression. NDV has a broad infectivity range for human cancer cells[23–25], and has been demonstrated to possess significant lytic activity in human mesothelioma, melanoma, glioblastoma, pancreatic, gastric, colon, and head and neck cancers. Consistent with these findings, infection of human melanoma cell lines with NDV-ICOSL resulted in 40- to over 400-fold induction of ICOSL on the surface of infected cells at 24 h (Fig. 2b). The induction of ICOSL expression, however, was significantly lower in most of the murine syngeneic tumour cell lines tested, including B16-F10 melanoma, MC38 and CT26 colon carcinoma, TRAMP C2 prostate carcinoma, ID8 ovarian carcinoma, 4T1 breast carcinoma, and MB49 bladder carcinoma (Fig. 2c). As the expression was most efficient in the B16-F10 cells, this cell line was selected as the primary model for the *in vivo* studies. Further *in vitro* characterization of the ICOSL-expressing virus revealed that the virus was equivalent to the control wild-type recombinant NDV (NDV-WT) with regard to its replicative capacity (Fig. 2d) and lytic ability (Fig. 2e) in B16-F10 cells. Intratumoral administration of NDV-ICOSL into B16-F10 tumours resulted in ICOSL upregulation in ∼2% of tumour cells at 24 h (Fig. 2f), which was consistent with the number of tumour cells predicted to be infected after the first cycle of viral replication (18 h) due to initial infection primarily taking place at the tumour periphery. To determine whether NDV infection could achieve sustained levels of transgene expression within B16-F10 tumours, we used recombinant NDV-expressing firefly luciferase (NDV-Fluc). Intratumoral injection of NDV-Fluc every 2 days into animals bearing B16-F10 tumours resulted in a strong luminescence signal, with augmentation following each injection, suggesting that expression of a therapeutic transgene *in vivo* would be feasible and could be sustained with repeated NDV administrations (Supplementary Fig. 4a–c).

**NDV-ICOSL enhances tumour control and TIL infiltration**. To evaluate NDV-ICOSL for therapeutic efficacy, animals bearing bilateral B16-F10 tumours were treated with four consecutive NDV intratumoral injections in one flank. As previously, we were unable to detect any virus in the contralateral tumours[13]. Both NDV-ICOSL and wild-type NDV were comparable in their ability to cause tumour regression within the tumours directly injected with the virus (Fig. 3a). However, when compared with the wild-type NDV, NDV-ICOSL resulted in significant tumour growth delay of the distant non-injected tumours with several animals remaining tumour-free long term (Fig. 3b,c). Analysis of virus-injected tumours revealed enhanced tumour infiltration with CD4[+]FoxP3[−] and CD8[+] cells in the

animals treated with NDV-ICOSL, which reached statistical significance over the NDV-WT group in the distant tumours (Fig. 3d). In addition, NDV treatment led to a small increase in the absolute number of Tregs, with the highest increase seen in the NDV-ICOSL group (Fig. 3d), although the relative percentage of Tregs was similar between the NDV-WT and NDV-ICOSL groups and significantly lower than in the PBS-treated group (Fig. 3e). In the treated tumours, NDV therapy led to expansion of different CD4[+]FoxP3[−] T-helper subsets defined by their lineage-defining transcription factors, including EOMES[+], GATA-3[+], Bcl-6[+], RORγt[+] and Tbet[+] lymphocytes, with the most significant increase seen in the latter population, defining the Th1 subset (Fig. 3f). The increase in each subset appeared to be augmented in the NDV-ICOSL treatment group, with the most significant increase seen in the Bcl-6[+] and RORγt[+] populations, defining the Tfh and Th17 subsets, respectively, while the expansion of the GATA-3[+] Th2 subset appeared to be dampened in the NDV-ICOSL group (Fig. 3f; Supplementary Fig. 5). Given the known role of ICOS in the development of follicular helper T cells, tumour-infiltrating lymphocytes (TILs) from the virus-treated tumours were characterized for the expression of Tfh lineage-specific markers (CD4[+]FoxP3[−]CXCR5[+]PD-1[+]ICOS[+]). There was an increase in the absolute number of Tfh cells, which was most pronounced in the NDV-ICOSL group, although the relative percentages of these cells out of all CD4[+]FoxP3[−] cells were very similar between the two groups (Fig. 3g). In summary, these data suggest that NDV-ICOSL may non-preferentially stimulate the expansion of several Th subsets, including the lymphocytes of Tfh, and Th17 lineages, while possibly having a negative effect on the expansion of the Th2 lymphocytes. Since these subsets represent a minority of the tumour-infiltrating CD4[+]FoxP3[−] cells, their exact role in the anti-tumour response is unclear, though they may serve to enhance the proliferation and activation of the tumour-infiltrating cytotoxic T cells. In support of this hypothesis, phenotypic characterization of the tumour-infiltrating CD8[+] cells from both treated and distant tumours revealed a more robust upregulation of the markers indicative of effector function (that is, ICOS and Granzyme B; Fig. 3h). There was a concomitant decrease of PD-1 expression in the CD8[+] lymphocytes in the treated tumours, with the most pronounced decrease seen in the NDV-ICOSL group (Fig. 3i). The downregulation of PD-1 coincided with reduction in PD-1[+] Granzyme B[−] CD8 cells and expansion of PD-1[−] Granzyme B[+] CD8 cells with NDV-ICOSL treatment (Fig. 3j). Overall, these findings demonstrate that intratumoral ICOS expression from NDV leads to enhanced abscopal therapeutic effect, with an associated intratumoral expansion of activated CD8[+] cells. This effect could be potentially mediated through direct stimulation of the CD8[+] cells by ICOSL, or indirectly through the expansion of the Th lineages noted above.

**NDV-ICOSL potentiates the efficacy of CTLA-4 blockade**. Overall, despite the significant inflammatory response seen in both virus-injected and distant tumours after intratumoral administration of NDV-ICOSL, the majority of the animals still succumbed to tumours, suggesting that active inhibitory mechanisms within the tumour microenvironment prevent tumour rejection by the infiltrating immune cells. Analysis of TILs from distant tumours revealed upregulation of CTLA-4 on the infiltrating Tcon and CD8[+] cells in both NDV- and NDV-ICOSL-treated animals, although there was no major difference between the two viruses (Fig. 4a). We thus proceeded to evaluate the efficacy of combination therapy using localized NDV-WT or NDV-ICOSL with systemic CTLA-4 blockade. For

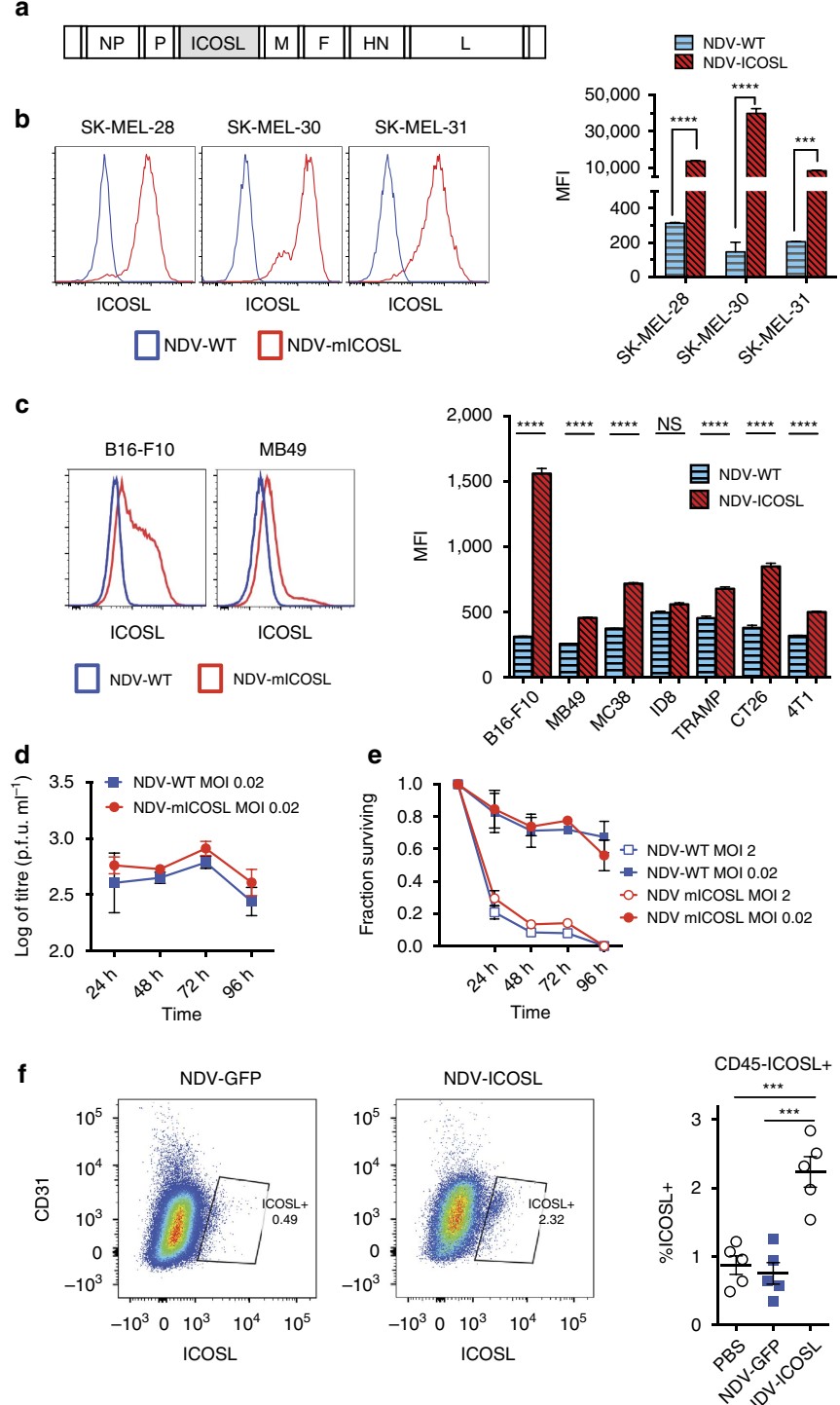

**Figure 2 | Generation of NDV-expressing murine ICOSL (NDV-ICOSL).** (**a**) NDV genomic construct scheme. (**b**) Expression of mICOSL in human melanoma cell lines 24 h after infection. Representative histograms (left) and median fluorescence intensity (MFI) bar graphs (right) are shown. (**c**) Expression of mICOSL on the surface of murine tumour cells 24 h after infection. Representative histograms from B16-F10 and MB49 cells (left), and MFI bar graphs (right) are shown. (**d**) Replication of parental NDV (NDV-WT) and NDV-ICOSL in B16-F10 cells. (**e**) *In vitro* lysis of B16-F10 cells by NDV-WT and NDV-ICOSL. (**f**) Expression of ICOSL in tumour cells 24 h after NDV-ICOSL treatment. Representative flow cytometry plots gated on CD45⁻ cells (left) and summary graph (right) are shown. ***$P < 0.001$, ****$P < 0.0001$, NS, non-significant. (**a–e**) Representative data from three experiments with three samples per group. (**f**) Representative data from two experiments with five animals per group. Data with error bars represent mean ± s.e.m. Two-tailed unpaired Student's *t* test was used for figures 2b,c, and one way ANOVA was used for 2f.

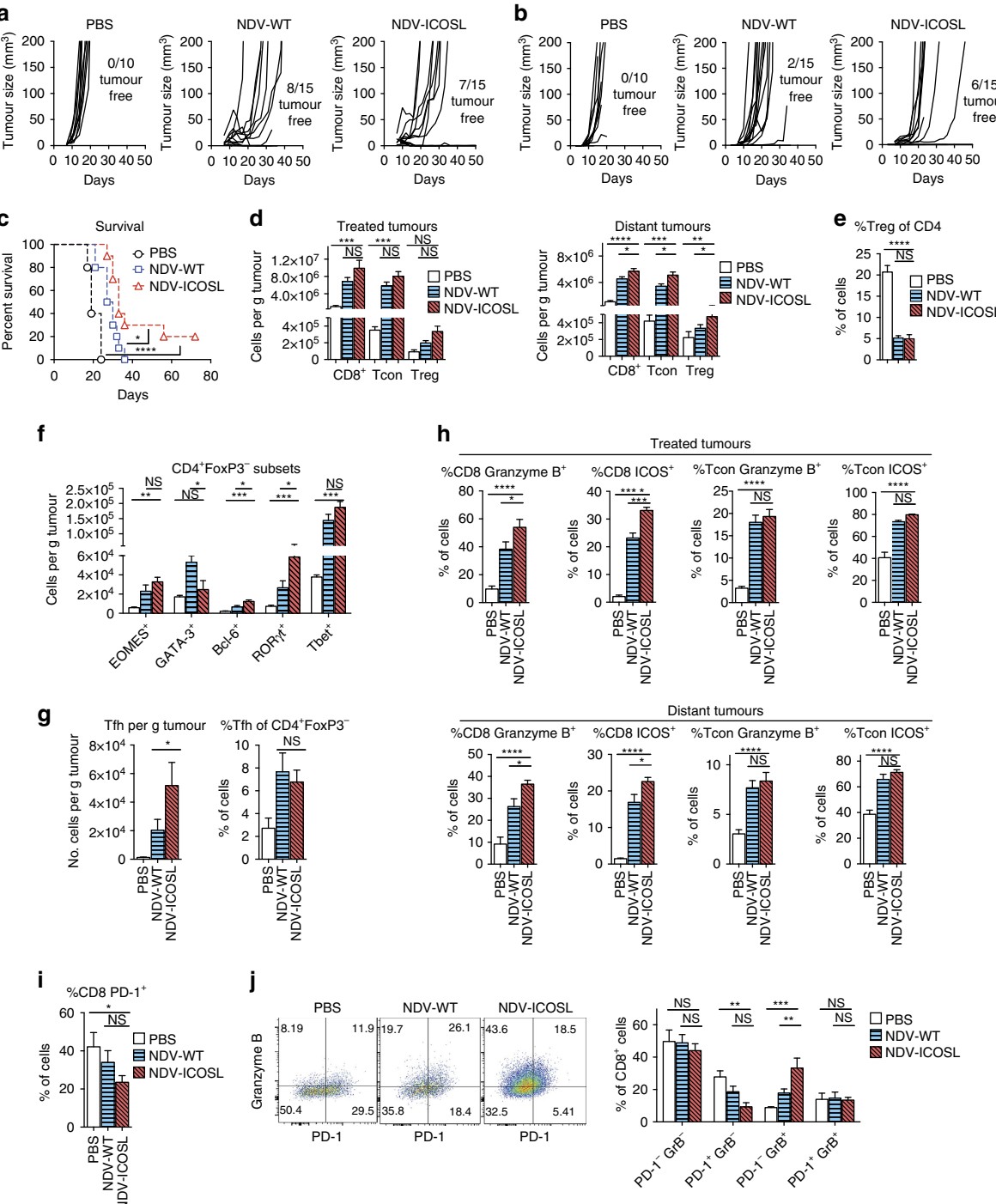

**Figure 3 | NDV-ICOSL improves tumour control and enhances tumour immune infiltration.** Bilateral flank B16-F10 tumour-bearing mice were injected into the right tumour with the indicated NDV. (**a**) Growth of individual virus-treated tumours. (**b**) Growth of distant tumours. (**c**) Overall survival. (**d**) Infiltration of the treated and distant tumours with different lymphocyte subsets. (**e**) Relative percentage of regulatory T cells in distant tumours. (**f**) Absolute numbers of EOMES$^+$, GATA-3$^+$, Bcl-6$^+$, ROR$\gamma$t$^+$ and Tbet$^+$ CD4$^+$FoxP3$^-$ lymphocytes in the treated tumours. (**g**) Absolute numbers of Tfh (CD4$^+$FoxP3$^-$CXCR5$^+$PD-1$^+$ICOS$^+$, left) and their percentages of CD4$^+$FoxP3$^-$ lymphocytes (right) in the treated tumours. (**h**) Expression of ICOS and Granzyme B (GrB) by tumour infiltrating CD8 and Tcon cells in the treated and distant tumours. (**i**) Expression of PD-1 by the CD8$^+$ lymphocytes in the treated tumours. (**j**) Percentages of CD8$^+$ lymphocytes double-stained for PD-1 and GrB expression. Left: representative flow cytometry plots; right: overall quantification. *$P < 0.05$, **$P < 0.01$, ***$P < 0.001$, ****$P < 0.0001$, NS, non-significant. (**a–c**) Pooled data of two experiments with 5–10 animals per group. (**d–e**) Representative data from two experiments with 10 animals per group. Data with error bars represent mean ± s.e.m. Log-rank test was used in 3c, one way ANOVA was used for 3d–j.

these experiments, we used the animals with a larger tumour challenge, where tumour control with combination of NDV-WT and CTLA-4 blockade was found to be suboptimal (Supplementary Fig. 1). The animals were treated with four

consecutive doses of NDV administered to a unilateral tumour, concurrently with systemic anti-CTLA-4 antibody as depicted in Fig. 4b. Combination therapy with NDV-ICOSL and anti-CTLA-4 led to regression of the majority of the virus-injected and

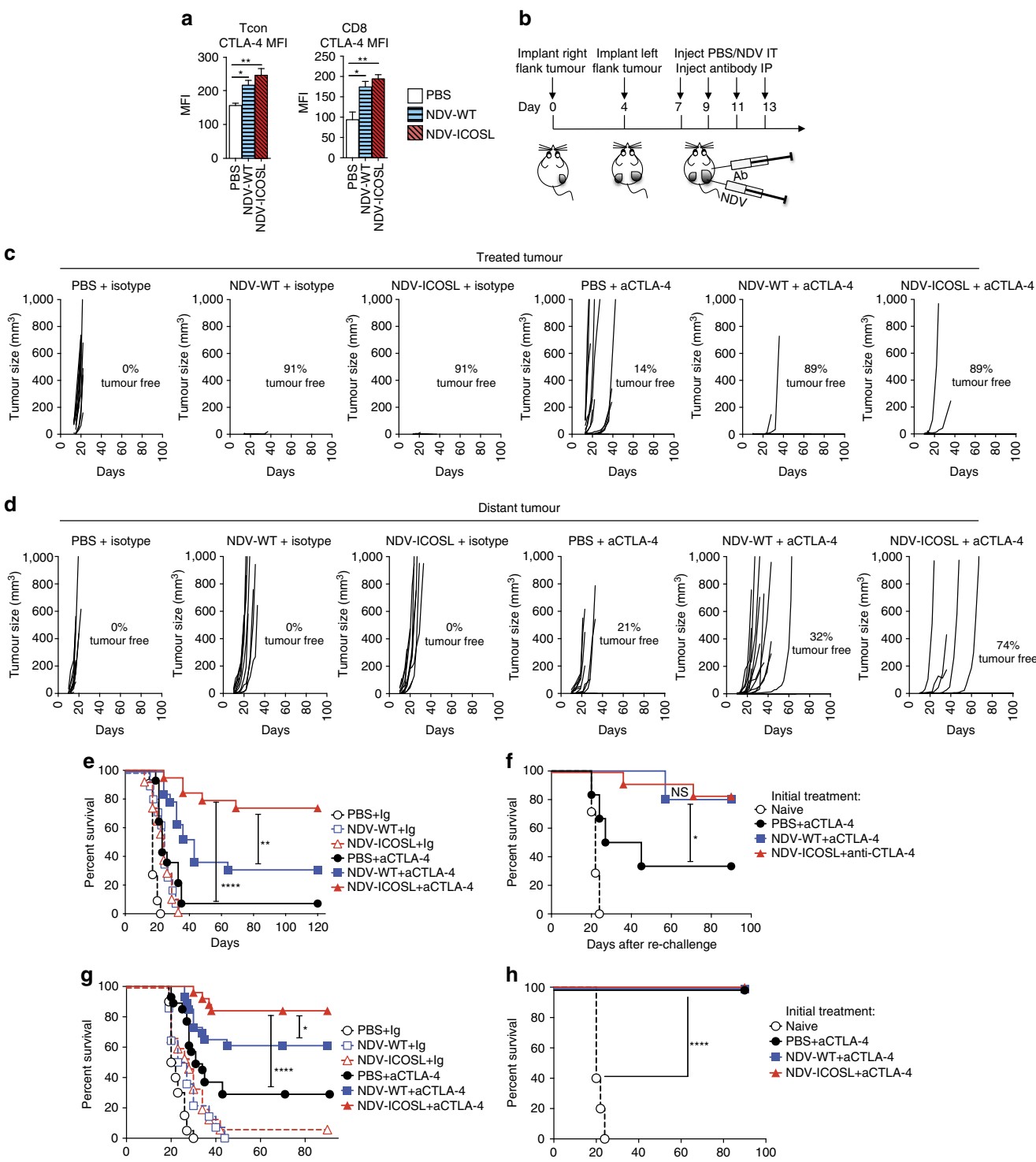

**Figure 4 | NDV-ICOSL potentiates the efficacy of CTLA-4 blockade. (a)** Upregulation of CTLA-4 in CD8[+] and Tcon lymphocytes in distant tumours. Median fluorescence intensity of CTLA-4 staining intensity is shown. (**b**) Treatment schema for combination therapy of NDV with CTLA-4 blockade in B16-F10 model with a larger tumour challenge ($1 \times 10^5$ cells in each flank). (**c**) Growth of individual NDV-injected B16-F10 tumours. (**d**) Growth of individual distant B16-F10 tumours. (**e**) Overall survival in the B16-F10 model. (**f**) Survival of B16-F10 tumour re-challenge at 90 days with no further treatment. (**g**) Overall survival of combination therapy in the CT26 model. (**h**) Survival of CT26 tumour re-challenge at 90 days with no further treatment. *$P < 0.05$, **$P < 0.01$, ****$P < 0.0001$, NS, non-significant. (**a**) Representative data of two experiments with five animals per group. (**c–e**) Pooled data from three experiments with 5–10 animals per group. (**f**) Pooled data from five experiments with final number of 5–12 animals per group. (**g,h**) Pooled data from three experiments with 5–10 animals per group. Data with error bars represent mean ± s.e.m. One way ANOVA was used for 4a,b and log-rank test was used for 4e–h.

distant tumours with long-term animal survival, which was superior to the combination of NDV-WT with anti-CTLA-4 (Fig. 4b–e). Animals that completely cleared tumours were re-implanted with tumour cells on day 90 without further therapy. We observed complete protection against re-implanted tumours in 80% of the animals. This protection was significantly greater in

the combination groups compared with that of the anti-CTLA-4-only group (Fig. 4f), indicating that the combination approach led to a more effective protective memory response. To validate these findings in other tumour models, we used the CT26 colon carcinoma and the MB49 bladder carcinoma models, although these cell lines exhibited significantly lower levels of ICOSL expression resulting from the poor infectivity described above (Fig. 2d). In the CT26 model, combination of NDV-ICOSL with CTLA-4 blockade achieved superiority over NDV-WT with CTLA-4, which was statistically significant (Fig. 4g), though the absolute magnitude of the benefit was lower than that seen in the B16-F10 model (Fig. 4e). All of the surviving animals were protected from tumour cell re-challenge (Fig. 4h), highlighting the high immunogenicity of this tumour model. In the MB49 model, where only minimal upregulation of ICOSL was seen (Fig. 2d), we observed no statistically significant improvement in survival, when NDV-ICOSL was compared with the parental NDV-WT virus in combination with CTLA-4 (Supplementary Fig. 6).

To evaluate the role of ICOS in NDV-based immunotherapy, a combination of NDV-WT with CTLA-4 blockade was initially evaluated in ICOS-deficient mice, using a small tumour challenge for which this combination was found to be effective. Surprisingly, while ICOS was previously found to be essential for the optimal efficacy of CTLA-4 blockade[26], ICOS deficiency did not affect the efficacy of the NDV-WT/anti-CTLA-4 combination (Supplementary Fig. 7a,b), suggesting that therapy with NDV-WT may upregulate additional pathways that could potentially compensate for the ICOS deficiency. These findings, however, did not exclude the role of ICOS in NDV-ICOSL therapy. To evaluate whether the therapeutic enhancement seen with NDV-ICOSL virus was mediated through the ICOS pathway, the combination of NDV-ICOSL/anti-CTLA-4 was also evaluated in ICOS$^{-/-}$ mice using a larger tumour challenge (Supplementary Fig. 7c,d). Therapeutic efficacy of the NDV-ICOSL/anti-CTLA-4 combination became attenuated in ICOS$^{-/-}$ mice, with overall survival mirroring that of NDV-WT/anti-CTLA-4 combination. These findings thus suggest that despite the apparently dispensable role of ICOS in the efficacy of NDV-WT/anti-CTLA-4 combination, provision of the immunostimulatory ICOSL signal by NDV in the tumour could drive therapeutic enhancement, an effect that was dependent on an intact ICOS signalling pathway. These findings support the previous studies using ICOSL cellular vaccine, demonstrating that therapeutic ICOSL solely acted through the ICOS pathway[21].

**Combination therapy leads to the expansion of activated TILs**. Analysis of distant B16-F10 tumours from the animals treated with combination of NDV and anti-CTLA-4 therapy demonstrated tumour infiltration with various immune cell subtypes (Fig. 5). The infiltration spanned both the innate (Fig. 5c,d) and the adaptive (Fig. 5e) immune compartments. While there were no significant differences in natural killer (NK) and myeloid cells between NDV-WT and NDV-ICOSL in the CTLA-4 combination groups, we observed a prominent increase of CD8$^+$ T cells in the group treated with combination of NDV-ICOSL and anti-CTLA-4. This increase was accompanied by the highest levels of the activation, lytic and proliferation makers—ICOS, granzyme B and Ki-67, respectively (Fig. 5g–i). Interestingly, there was also a small increase in Tregs (Fig. 5e), though the overall percentage of Tregs was significantly decreased when compared with the untreated animals or animals treated with single-agent anti-CTLA-4 (Fig. 5f). Re-stimulation of the isolated TILs with DCs loaded with B16-F10 tumour lysates demonstrated an increase in IFN$\gamma$ production by intracellular cytokine staining suggestive of

tumour-specific immunity, which was most pronounced in combination therapy groups, although there was no significant difference between NDV-WT and NDV-ICOSL (Fig. 5j; Supplementary Fig. 8).

To examine the changes occurring with the combination therapy on both local and systemic level, we analysed the effect of NDV-ICOSL and CTLA-4 blockade on the TILs from the treated tumours and on the splenic T-cell populations (Fig. 6; Supplementary Figs 9 and 10).

In the treated tumours, combination therapy of NDV-ICOSL with CTLA-4 blockade resulted in the most marked increase in CD8$^+$ and Tcon cells (Fig. 6a). There appeared to be a concomitant increase in the absolute Tfh numbers in these tumours (defined by CD4$^+$FoxP3$^-$CXCR5$^+$PD-1$^+$ICOS$^+$ lineage), although the relative percentage of these cells remained low and not substantially different from the other treatment groups (Fig. 6b).

In the spleen, treatment with either combination had no substantial effect on the spleen size or absolute numbers of the splenic CD8$^+$, Tcon and Treg populations (Supplementary Fig. 9). Both NDV-WT and NDV-ICOSL viruses led to expansion of the splenic Tfh populations, defined by expression of Bcl-6 or by CD4$^+$FoxP3$^-$CXCR5$^+$PD-1$^+$ICOS$^+$ lineage markers (Supplementary Fig. 10). While CTLA-4 blockade augmented the expansion of Tfh cells, supporting prior findings, demonstrating the regulatory role of CTLA-4 in Tfh responses[27], there were no substantial differences seen between the NDV-WT and NDV-ICOSL viruses. In both treated tumours and the spleen, NDV-ICOSL/anti-CTLA-4 treatment resulted in the highest increase in the percentages of CD8$^+$ cells characterized by high expression of Granzyme B and ICOS (Fig. 6c–f). Overall, these demonstrate that treatment with NDV-ICOSL with CTLA-4 blockade is associated with expansion and activation of CD8$^+$ lymphocytes in the treated tumours, spleen and distant tumours, suggesting that this population of cells may be responsible for the observed increase in therapeutic efficacy.

**Discussion**

Immunotherapeutic strategies have shown significant promise for treatment of cancers resistant to conventional modalities, leading to Food and Drug Administration approval of agents targeting the CTLA-4 and PD-1 pathways. Clinical studies have already demonstrated an advantage from combined CTLA-4 and PD-1 blockade in melanoma, which must be considered in the context of increased adverse event frequency[7].

Despite such clinical results, even with combined checkpoint blockade, therapeutic success has so far been limited to a subset of patients, calling for identification of markers predicting response and development of combinatorial therapeutic approaches. In addition, the increased toxicity from systemic immune activation highlights the rationale for exploration of approaches targeting the immune system in a locoregional manner. A rational approach towards design of a combinatorial regimen would ideally focus on several aspects, including enhancement of antigen presentation, improvement of T-cell effector function and inhibition of immunological checkpoints[28].

To this end, locoregional/intratumoral therapy with OVs presents an attractive strategy, as it achieves tumour lysis, immunogenic cell death and production of type I IFN, creating a microenvironment favourable for the activation of anti-tumour immune response[29]. Genetically engineered replicative OVs provide for an additional opportunity for improved intratumoral spread and delivery of danger signals to the innate immune effectors, as well as intratumoral immunomodulation through expression of specific therapeutic ligands. OVs

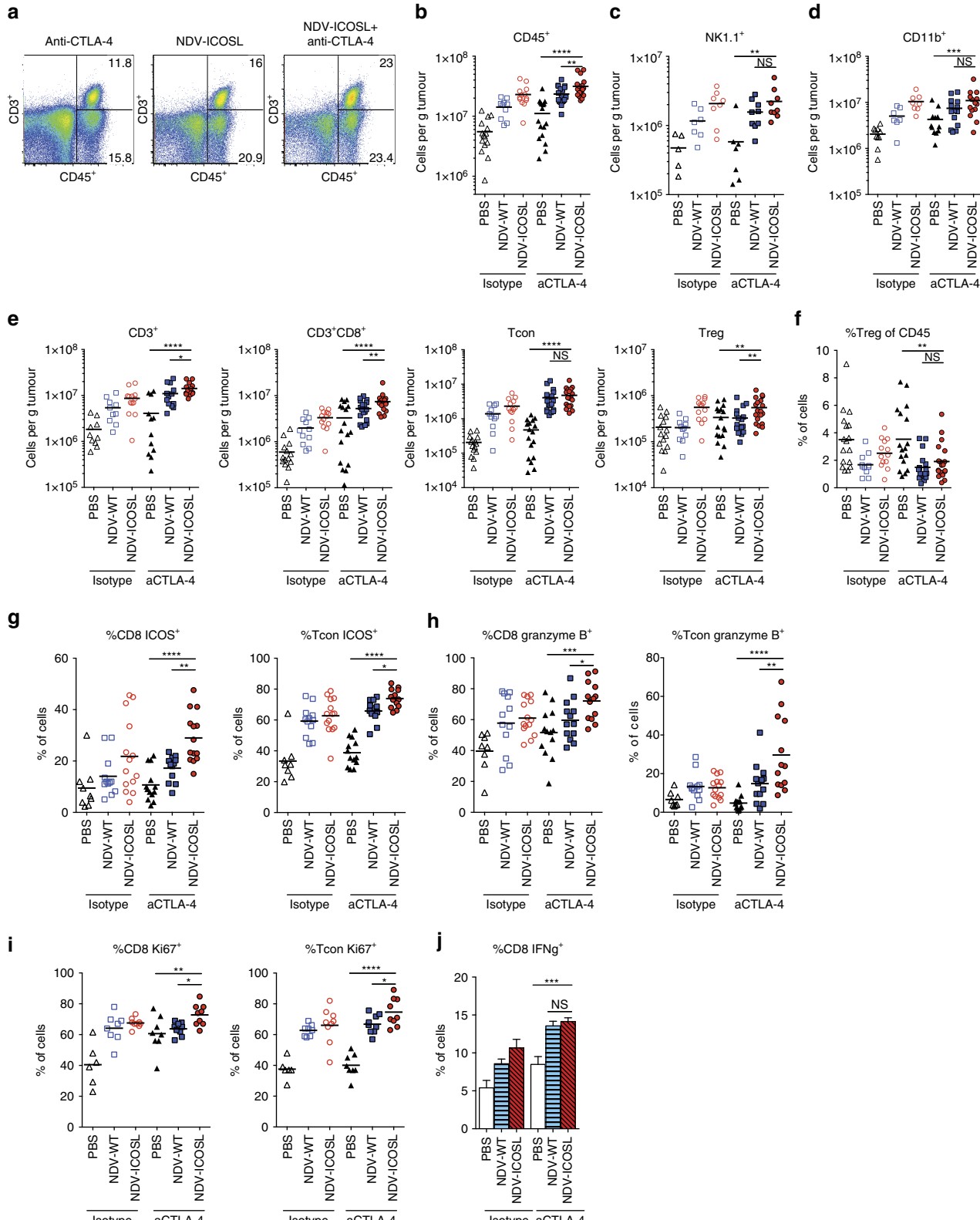

**Figure 5 | Combination therapy leads to expansion of activated TILs.** Animals were challenged with a large tumour cell dose ($2 \times 10^5$ cells in each flank) to allow tumour outgrowth, treated according to the schedule in Fig. 4b, and tumours were collected on day 15. (**a**) Representative flow cytometry plots of CD45$^+$CD3$^+$ cells gated on total live-cell population. (**b–d**) Absolute number of CD45$^+$ (**b**), CD11b$^+$ (**c**) and NK1.1$^+$ (**d**) cells per gram of tumour. (**e**) Absolute numbers of lymphocytes per gram of tumour. (**f**) Relative percentage of Tregs of CD45$^+$ cells. (**g–i**) Upregulation of ICOS (**g**), Granzyme B (**h**) and Ki-67 (**i**) on tumour-infiltrating CD8$^+$ and CD4$^+$FoxP3$^-$ lymphocytes. (**j**) Expression of IFNγ by tumour-infiltrating CD8$^+$ lymphocytes in response to re-stimulation with DC's loaded with tumour lysates. *$P < 0.05$, **$P < 0.01$, ***$P < 0.001$, ****$P < 0.0001$, NS, non-significant. Pooled data from three experiments with 5–10 animals per group. Data with error bars represent mean ± s.e.m. One way ANOVA was used for the analyses.

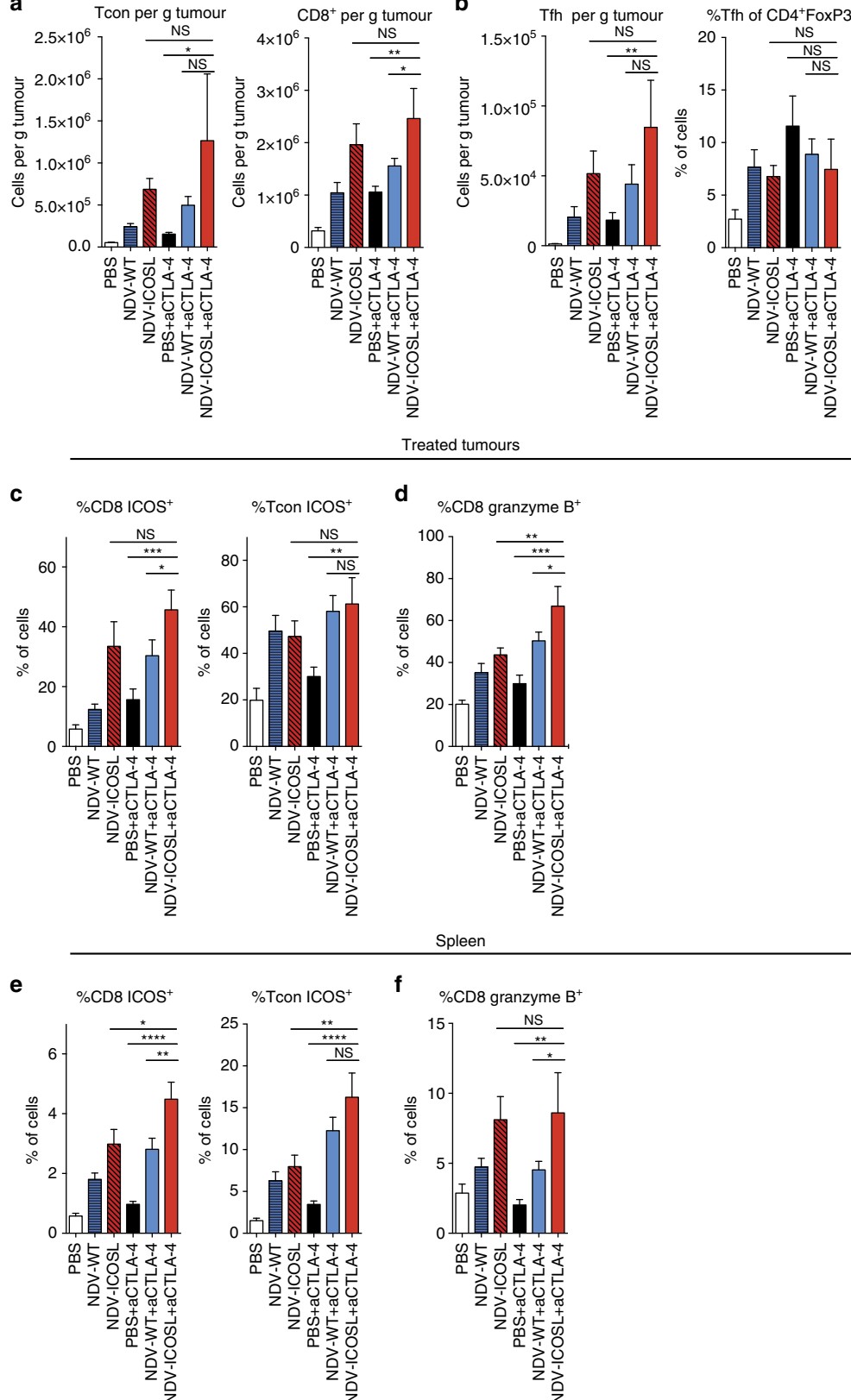

**Figure 6 | NDV-ICOSL enhances local and systemic CD8 T-cell activation.** Animals were treated as in Fig. 5 and the virus-infected tumours and spleens were collected on day 15. (**a**) Absolute numbers of Tcon and CD8[+] cells in the treated tumours. (**b**) Absolute numbers and relative percentages of the Tfh (CD4[+]FoxP3[−]PD-1[+]CXCR5[+]ICOS[+]) in the treated tumours. (**c**) Expression of ICOS by the Tcon and CD8[+] cells in the treated tumours. (**d**) Expression of Granzyme B by the tumour-infiltrating CD8 lymphocytes. (**e**) Expression of ICOS by the splenic lymphocytes. (**f**) Expression of Granzyme B by the splenic CD8 lymphocytes. Representative data from two experiments with five animals per group. Data with error bars represent mean ± s.e.m. One-way ANOVA was used for the analyses.

engineered to enhance innate immune responses through the production of GM-CSF are currently the most advanced in clinical development[19,20], with the recent Food and Drug Administration approval of intralesional talimogene laherparepvec for the treatment of metastatic melanoma. Despite these advances, the appropriate targets for intratumoral immuno-modulation with OVs are still not known, and are highly influenced by the individual virus biology, replicative capacity and the interaction of the virus with the tumour and stromal cells[10].

We have previously demonstrated that intralesional therapy with attenuated NDV could potentiate the efficacy of systemic CTLA-4 blockade[13]. The immunostimulatory activity of NDV is thought to be mediated through the induction of type I IFN and chemokines, upregulation of major histocompatibility complex and cell adhesion molecules, and facilitated adhesion of lymphocytes and APC's through the expression of viral glycoproteins on the surface of infected cells[30–32]. As this virus appears to work well with immune checkpoint blockade, in the current study we explored locoregional pathways that could be further targeted with the virus to achieve improved systemic immunotherapeutic efficacy. To this end, a CD28 homologue ICOS was identified as one of the molecules strongly upregulated in response to NDV therapy. Signalling through ICOS induces T-cell activation and proliferation, and was shown to be critical for T-cell-dependent B lymphocyte responses and the development of T-helper subsets, including Th1, Th2, Th17 and Tfh (refs 33–41). In a neoadjuvant trial for bladder carcinoma with ipilimumab, upregulation of ICOS by TILs as well as circulating lymphocytes was highly correlated with response to therapy[42]. The isolated ICOS$^+$ effector T cells were capable of recognizing tumour antigens and secreting IFN-$\gamma$ (ref. 42). Similarly, a phase I clinical trial with anti-CTLA-4 therapy in patients with metastatic hormone-responsive breast cancer revealed a significant increase in the percentage of ICOS$^+$ peripheral blood mononuclear cells[43]. Finally, a retrospective analysis of cancer patients treated for metastatic melanoma revealed a sustained increase in the frequency of ICOS$^+$ T cells in patients who derived clinical benefit from therapy[44]. The role of ICOS in anti-tumour tumour efficacy of CTLA-4 blockade was confirmed by mouse studies, where therapeutic efficacy of CTLA-4 blockade was severely compromised in ICOS-deficient mice[26]. Surprisingly, in our study, combination therapy of NDV with CTLA-4 blockade was not diminished in the ICOS$^{-/-}$ mice. These findings suggest that, in the setting of NDV infection, activation of additional T-cell pathways may compensate for the ICOS deficiency. Indeed, in response to NDV, we observed upregulation of co-stimulatory members of both immunoglobulin and TNF receptor superfamily, including CD28, 4-1BB, GITR and OX40 (Fig. 1h).

These findings, however, did not exclude a possibility that additional signalling through the upregulated ICOS target in the tumour microenvironment could further enhance therapeutic efficacy of NDV. Indeed, when tested in vivo, recombinant NDV-expressing ICOSL demonstrated therapeutic superiority over the parental NDV virus, with enhanced abscopal effect in distant tumours (Fig. 3). This effect was associated with increases in absolute numbers of several intratumoral Th subsets, including the lymphocytes of Tfh, and Th17 lineages, with potential inhibitory effect on the expansion of the Th2 lineage. Since these subsets represent a minority of the tumour-infiltrating CD4$^+$ FoxP3$^-$ cells, their exact role in the anti-tumour response is unclear. The most pronounced differences, however, were seen in the expansion and activation of the intratumoral CD8$^+$ lymphocytes (Fig. 3h), an effect that was further augmented by combination with systemic CTLA-4 blockade and was seen in the

virus-injected tumours, distant tumours and spleen (Figs 5 and 6). We speculate that CD8 expansion and activation could be mediated through two mechanisms: direct, through sustained ICOS signalling in the post-priming phase through provision of the ICOSL to the CD8 lymphocytes, or indirect, through increase in the intratumoral Th sublineages such as Tfh and Th17 cells. Interestingly, the expression of ICOSL by NDV appeared to be insufficient to drive anti-tumour effect at the injected tumour site, which could be potentially explained by differential regulation by ICOSL of virus versus tumour-specific TILs. For example, prior studies with vesicular stomatitis virus (VSV)-expressing CD40L demonstrated that CD40L interfered with the efficacy of VSV by shifting the immune response to predominantly being anti-viral rather than anti-tumour[45]. Nevertheless, this finding suggests that studies of immunotherapeutic efficacy of recombinant OVs solely on the basis of anti-tumour activity in the injected tumours could be misleading. The abscopal effect seen with OVs provides a better measure of systemic anti-tumour immunity, and is more likely to be therapeutically relevant. To this end, the combination of NDV-ICOSL with systemic CTLA-4 blockade led to the rejection of virus-injected and distant tumours and long-term survival in the majority of animals (Fig. 4). The observed increase in therapeutic efficacy was thus likely mediated by enhanced activation of lymphocytes through ICOS signalling mediated at the virus-treated tumour site, with resultant enhancement in the abscopal function. These results mirror previous findings from our group, demonstrating that a cellular tumour vaccine expressing ICOSL was able to significantly enhance the efficacy of systemic CTLA-4 blockade, an effect that was associated with expansion and enhancement of the effector function of the tumour-infiltrating CD8$^+$ lymphocytes[21]. We cannot fully exclude that ICOSL could still be acting through additional unidentified pathways, which may be more apparent in a different model. For example, a study by Yao et al., has demonstrated that in addition to ICOS, ICOSL has CD28 and CTLA-4 as potential partners, although this was only observed with human and not mouse molecules[46]. This, however, does not exclude a possibility that mouse ICOSL could have additional interaction partners.

Interestingly, NDV treatment also increased the absolute numbers of CD4$^+$FOXP3$^+$ cells, which was more pronounced in the NDV-ICOSL-treated group. This could be a compensatory response to the increase in the conventional and CD8$^+$ T-cell populations, as the tumours of the treated animals with the highest numbers of CD4$^+$FoxP3$^+$ cells also exhibited the highest numbers of the conventional CD4$^+$FoxP3$^-$ T cells and CD8$^+$ cells. However, the increase in Tregs in response to ICOS agonism could also be a reflection of a known important role of ICOS signalling in Treg development and homeostasis[36,39]. A recent study by Metzger et al.[47] demonstrated that, in animal tumour models, ICOSL blockade resulted in reduction of intratumoral Tregs. Despite these findings, ICOSL blockade resulted in reduced therapeutic efficacy of OX40 agonistic antibody, likely due to the impairment of ICOS-dependent CD4 effector responses. Conversely, expression of ICOSL by tumour cells promoted OX40 agonist-mediated tumour rejection and survival. These findings mirror our results, demonstrating a significant improvement in therapeutic efficacy seen with NDV-ICOSL, likely explained by the preferential effect of ICOSL on expansion and effector function of the tumour-infiltrating CD8$^+$ and conventional CD4$^+$ T cells. One could, however, hypothesize that the therapeutic effect might be further optimized through the utilization of additional strategies that deplete Tregs.

The expression of a co-stimulatory ligand from a virus within the tumour may present an advantage over systemic administration, primarily with regards to avoiding systemic toxicity. Our

studies indicate that with intratumoral administration of NDV, the expression of ICOSL is limited to only a small percentage of cells and is likely short-lived (Fig. 2f; Supplementary Fig. 4), though this appears to be sufficient to enhance anti-tumour effect. NDV exhibits a broad infectivity for human cancer cell lines and can lead to high levels of therapeutic transgene expression in the infected human cancer cells (Fig. 2b)[23,24]. The infectivity of NDV for mouse cell lines, however, appears to be lower, with B16-F10 cell line supporting the highest levels of transgene expression (Fig. 2c), though even in this cell line the overall level of ICOSL expression was at least an order of magnitude lower than that seen in human cancer cells (Fig. 2b). The levels of ICOSL transgene expression from NDV in mouse tumour cell lines directly correlated with NDV-ICOSL efficacy in these tumour models, with the best effect seen in the B16-F10 and CT26 models, and virtually no enhancement seen in the MB49 model (Fig. 4; Supplementary Fig. 5). Given the higher infectivity of NDV in human cancer cells, this finding might not be relevant, though exploration of additional strategies aiming to enhance the levels of therapeutic ligand expression in tumours may be warranted.

In summary, these findings serve as a proof of principle that locoregional immunotherapy with oncolytic virus may upregulate negative (for example, CTLA-4) and positive (for example, ICOS) immune feedback mechanisms, and that targeting of these mechanisms through both systemic and locoregional approaches may be required for optimal therapeutic efficacy. We demonstrate that ICOS is an attractive target for locoregional immunotherapy with NDV and provide a strong rationale for clinical evaluation of therapeutic approaches targeting ICOS within the tumour microenvironment, such as the one described in our study.

## Methods

**Study design.** The primary research objective was to define co-stimulatory molecules upregulated in the tumour microenvironment by NDV and to design strategies for intratumoral targeting of such molecules in combination with systemic CTLA-4 blockade. The pre-specified hypothesis suggested that NDV infection would induce upregulation of pro-inflammatory pathways, which could be further targeted through additional co-stimulation. The overall study design was a series of controlled laboratory experiments in mice, as described in the sections below. In all of the studies, the assignment of animals to experimental groups was random. For survival studies, sample sizes of 5–10 mice per group were used and pooled data from several replicate experiments were used for statistics. With 20 mice per group, 90% power and a 5% significance level, we could detect differences in tumour-free survival from 40 to 70%. Survival analyses were performed using the log-rank test. The experiments were replicated two to three times as noted and the final analysis included either pooled data or representative experiments where indicated. For the experiments reporting isolation of TILs, five mice per group were used for each experiment, with three to five replicates. All outliers were included in the data analysis.

**Mice.** C57BL/6 and C57BL/6 ICOS$^{-/-}$ mice were purchased from Jackson Laboratory. All mice were maintained in microisolator cages and treated in accordance with the National Institute of Health and American Association of Laboratory Animal Care regulations. All mouse procedures and experiments in this study were approved by the Memorial Sloan Kettering Cancer Center Institutional Animal Care and Use Committee.

**Animal imaging.** Mice were imaged starting at 24 h after initial NDV injection. Mice were injected retro-orbitally with 50 μl of 40 mg ml$^{-1}$ D-luciferin (Caliper Life Sciences) in PBS and imaged immediately using the IVIS Imaging System (Caliper Life Sciences). Gray-scale photographic images and bioluminescence colour images were superimposed using The Living Image, version 4.0 (Caliper Life Sciences) software overlay. A region of interest was manually selected over the tumour and the area of the region of interest was kept constant.

**Cell lines.** The murine cancer cell lines for melanoma (B16-F10, originally provided by Dr Isaiah Fidler, MD Anderson Cancer Center, Houston, TX), colon cancer (CT26 and MC38, originally obtained from American Type Culture Collection) and bladder carcinoma (MB49, originally generated by Dr Ian

Summerhayes at Imperial Cancer Research Fund, UK) were maintained in RPMI medium supplemented with 10% fetal calf serum and penicillin with streptomycin. A549 cells obtained from American Type Culture Collection were cultured in DMEM medium, supplemented with 10% fetal calf serum and penicillin with streptomycin. Human melanoma cell lines, originally isolated from patients at MSKCC (SK-MEL-28, SK-MEL-30 and SK-MEL-31) were maintained in RPMI medium supplemented with 10% fetal calf serum and penicillin with streptomycin. All of the cell lines have been tested and were found to be negative for mycoplasma contamination. The cell lines have not been re-authenticated since their receipt from the original sources.

**Antibodies.** Therapeutic anti-CTLA-4 (clone 9H10) antibody and isotype control antibody was purchased from BioXcell (cat: BE0131 and BE0087). Antibodies used for flow cytometry were purchased from the following sources (dilutions are indicated in parentheses): eBioscience (CD45.2 Alexa Fluor 700, cat: 56-0454 (1:200), CD3 PE-Cy7, cat: 25-0031 (1:200), CD4 ef450, cat: 48-0041 (1:200), CD4 APC-efluor780, cat: 47-0041 (1:400), CD8 PerCP-efluor710, cat: 46-0083 (1:200), CD11b APC-efluor 780, cat: 47-0112 (1:600), ICOS PE, cat: 12-5985 (1:200), ICOSL PE, cat: 12-5985 (1:200), CTLA-4 PE, cat: 12-1522 (1:200), NK1.1 PE, cat: 12-5941 (1:200), IFNγ PE, cat: 12-7311 (1:200), FoxP3 Alexa Fluor 700, cat: 56-5773 (1:100), FoxP3 APC, cat: 17-5773 (1:200), GATA-3 PE, cat: 12-9966 (1:100), RORγT PerCP-efluor710, cat: 46-6981 (1:100), Tbet PE-Cy7, cat: 25-5825 (1:100), EOMES efluor 450, cat: 48-4875 (1:100), PD-1 PE-Cy7, cat: 25-9985, (1:200)), Biolegend (CD3 BV570, cat: 100225 (1:100), CD11b BV570, cat: 101233 (1:50), Bcl-6 Alexa 594, cat: 648308 (1:50)), Invitrogen (Granzyme B PE-Texas Red, cat: GRB17 (1:125), Granzyme B APC, cat: GRB05 (1:125)) and BD Pharmingen (Ki-67-Alexa Fluor 488, cat: 561165 (1:50), CXCR5-biotin, cat: 551960 (1:100)).

**Viruses.** Recombinant lentogenic NDV LaSota strain was used for all experiments. To generate NDV-expressing murine ICOSL, a DNA fragment encoding the murine ICOSL flanked by the appropriate NDV-specific RNA transcriptional signals was inserted into the SacII site created between the P and M genes of pT7NDV/LS. Viruses were rescued from complementary DNA using methods described previously[22] and sequenced by reverse transcription PCR for insert fidelity[22]. Rescued viruses were passaged in embryonated 9-day-old hen eggs. Virus titres were determined by serial dilution and immunofluorescence in A549 cells.

***In vitro* infection experiments.** For evaluation of surface ICOSL expression in cell lines, cells were infected at the indicated multiplicity of infection (MOI) for 24 h, and were processed 24 h later for surface antibody labelling and flow cytometry. For replication kinetics, 5e4 B16-F10 cells were infected in 12-well plates at MOI of 0.02, with an input virus of 1e3 p.f.u. in a total volume of 100 μl. After 1 h attachment, the infection media was aspirated, and the cells were incubated with 2 ml of fresh media at 37 °C in 1 ml of DMEM with 250 ng ml TPCK trypsin. Culture supernatants were collected at 24, 48 and 72 h, and virus titres were determined by serial dilution and immunofluorescence in A549 cells. For *in vitro* cytotoxicity experiments, infections were carried out in a similar manner, with B16-F10 cells infected in six-well dishes at 25% confluency. At 24, 48, 72 and 96 h post infection, cells were washed and incubated with 1% Triton X-100 at 37 °C for 30 min. Lactate dehydrogenase activity in the lysates was determined using the Promega CytoTox 96 assay kit, according to the manufacturer's instructions.

**Tumour challenge survival experiments.** Bilateral flank tumour models were established to monitor for therapeutic efficacy for both injected and systemic tumours. Treatment schedules and cell doses were established for each tumour model to achieve 10–20% tumour clearance by NDV or anti-CTLA-4 as single agents. For experiments comparing therapeutic efficacy of NDV-WT and NDV-ICOSL as single agents, B16-F10 tumours were implanted by injection of $2 \times 10^5$ B16-F10 cells in the right flank intradermally (i.d.) on day 0 and $5 \times 10^4$ cells in the left flank on day 4. On days 7, 10, 13 and 16, mice were treated with four 100 μl intratumoral injections of NDV in PBS ($2 \times 10^7$ p.f.u.). For combination experiments with CTLA-4 blockade in the B16-F10 model, bilateral flank tumours were established as above, with injection of $2 \times 10^5$ B16-F10 cells into the right flank i.d. on day 0 and $1 \times 10^5$ cells in the left flank on day 4. On days 7, 10, 13, and 16, mice were treated with four intratumoral injections of appropriate NDV, and concurrently received four intraperitoneal (i.p.) injections of anti-CTLA-4 antibody (100 μg). Control groups received a corresponding dose of isotype antibody i.p. and intratumoral injection of PBS. Tumour size and incidence were monitored over time by measurement with a caliper. Tumour volume was estimated using the following formula: length × width × height/2. On day 90, the surviving animals were rechallenged with $2 \times 10^5$ B16-F10 cells in the left flank and were monitored for survival. For the CT26 model, $1 \times 10^6$ cells were implanted in right flank on day 0 and left flank on day 2. Treatment was performed on days 6, 8 and 10 in a similar manner as above. For the MB49 model, $5 \times 10^5$ cells were implanted in right flank and left flank on day 0. Treatment was performed on days 7, 9, 11 and 13 in a similar manner as above.

**Isolation of TILs and splenocytes.** B16-F10 tumours were implanted by injection of $2 \times 10^5$ B16-F10 cells in the right flank i.d. on day 0 and $2 \times 10^5$ cells in the left flank on day 4. On days 7, 10 and 13, the mice were treated with three intratumoral injections of $2 \times 10^7$ p.f.u. of appropriate NDV, and 100 μg of i.p. anti-CTLA-4 antibody, where specified. On day 15, mice were killed by $CO_2$ inhalation. Tumours and spleens were removed using forceps and surgical scissors and weighed. Tumours from each group were minced with scissors before incubation with 1.67 Wünsch U ml$^{-1}$ Liberase and 0.2 mg ml$^{-1}$ DNase for 30 min at 37 °C. Spleens were subjected to red blood cell lysis using 1 ml of red blood cell lysis buffer (Sigma) for 1 min at room temperature. Tumours and spleens were homogenized by repeated pipetting and filtered through a 70-μm nylon filter. Cell suspensions were washed once with complete RPMI.

**Flow cytometry.** Cells isolated from tumours or tumour-draining lymph nodes were processed for surface labelling with several antibody panels staining CD45, CD3, CD4, CD8, CTLA-4, ICOS, CD11c, CD19, NK1.1, CD11b, PD-1, Ki-67, Tbet, RORγt, Bcl-6, GATA-3, EOMES and CXCR5. Fixable viability dye eFluor780 (eBioscience) was used to distinguish the live cells. Cells were further permeabilized using FoxP3 fixation and permeabilization kit (eBioscience), and stained for Ki-67, FoxP3, Granzyme B or IFNγ. Data were acquired using the LSRII Flow cytometer (BD Biosciences) and analysed using FlowJo software (Treestar). The quantifications of lymphocytes in all figures are based on CD45$^+$CD3$^+$CD11b$^-$ gate.

**DC Purification and loading.** Spleens from naive mice were isolated and digested with 1.67 Wünsch U ml$^{l-}$ Liberase (Roche) and 0.2 mg ml$^{-1}$ DNase (Roche) for 30 min at 37 °C. The resulting cell suspensions were filtered through 70-μm nylon filter and washed once with complete RPMI. CD11c$^+$ DCs were purified by positive selection using Miltenyi magnetic beads. Isolated DCs were cultured overnight with recombinant GM-CSF and B16-F10 or TRAMP C2 tumour lysates and were purified on Ficoll gradient.

**Analysis of cytokine production.** Cell suspensions from tumours were pooled and enriched for T cells using a Miltenyi T-cell purification kit. Isolated T cells were counted and co-cultured for 8 h with DCs loaded with B16-F10 or TRAMP C2 tumour cell lysates in the presence of 20 U ml$^{-1}$ IL-2 (R and D) plus Brefeldin A (BD Bioscience) in a 1:5 (DC: lymphocyte) ratio. After re-stimulation, lymphocytes were processed for flow cytometry as above.

**NanoString gene expression analyses.** B16-F10 tumours were implanted by injection of $4 \times 10^5$ B16-F10 cells into the right flank i.d. on day 0. On days 7, 10 and 13, the mice were treated with three intratumoral injections of NDV ($2 \times 10^7$ p.f.u.). On day 15, the animals were killed, and the tumours were excised, placed in TRIzol reagent (Invitrogen) and homogenized. The samples were flash-frozen in dry ice and ethanol and stored at $-80$ °C. RNA was later purified from TRIzol using the Direct-zol RNA MiniPrep kit (Zymo Research). Isolated RNA was hybridized with the NanoString nCounter PanCancer Immune Profiling mouse panel codeset and quantified using the nCounter Digital Analyzer at the MSKCC Genomics Core Facility. Data were processed nSolver Analysis Software, using the Advanced Analysis module.

**Statistics.** Statistical significance for all flow cytometric analyses was determined utilizing two-tailed unpaired Student's $t$-tests (for comparison of two groups) and one-way analysis of varinace (for multiple group comparisons) with $\alpha = 0.05$. For survival analyses, log-rank testing (Mantel–Cox) was performed. The numbers of animals included in the study are discussed in each figure.

**Data availability.** All relevant data are available from the corresponding author (D.Z.) upon reasonable request.

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

## Acknowledgements

This work was supported by HHMI (J.P.A.), the National Institutes of Health (CA056821 to J.D.W.), the Ludwig Institute for Cancer Research (J.D.W.) and Swim Across America (J.D.W.). D.Z. is the Bart A. Kamen Fellow of the Damon Runyon Cancer Research Foundation and received funding from the Bladder Cancer Awareness Network, ASCO Conquer Cancer Foundation and MSKCC Cycle for Survival. MSKCC Genomics Core Facility is supported by the NCI Core grant P30 CA008748.

## Author contributions

D.Z. designed and performed the experiments, analysed the data and prepared the manuscript. R.B.H, J.R. and T.P. performed the experiments and assisted in manuscript preparation. P.P., P.S., T.M., J.D.W. and J.P.A. assisted in experimental design, data interpretation and manuscript preparation.

## Additional information

**Competing financial interests:** J.P.A. is an inventor on patents concerning targeting CTLA-4 for cancer therapy, licensed to Bristol Myers Squibb. J.P.A. and P.S. are inventors on a patent concerning targeting of ICOS, licensed to Jounce Therapeutics. D.Z., P.P., J.D.W. and J.P.A. are inventors on a patent concerning the uses of recombinant Newcastle Disease Virus for cancer therapy. The remaining authors declare no competing financial interests.

