## [Peer Review File · Nature Communications]

Reviewers' comments:

Reviewer #1 (Remarks to the Author):

The title: Intratumoral modulation of the inducible costimulator (ICOS) by recombinant oncolytic virus promotes systemic anti-tumor immunity.

D. Zamarin et al. found that oncolytic NDV stimulates a robust increase in mDCs at tumor sites and identified that ICOS is significantly higher in TIL amongst all co-stimulation factors analyzed by nanostring. Based on this finding, the authors introduced a ICOS ligand cDNA in a recombinant NDV genome and treated bilateral tumors bearing mice with NDV-ICOSL: the treatment led to significant tumor rejection at a separate flank site when compared with parental NDV-wt and control groups. The authors identified more effector T cells in the TIL of distinct tumor mass. These findings are very interesting and important in immunotherapy using oncolytic viruses because a current limitation in the clinical setting is that therapeutic OV's can only be injected at some tumor sites and cannot be applied to all metastatic sites.

The authors should clarify the following issues:

1) ICOS is thought to be important for helper CD4 subsets (i.e. T follicular, Th1/2/17) during immunization. The authors also show increased CD4 and CD8 T cells at distant tumor sites when NDV-ICOSL was administered to the right flank tumor. However, they do not provide this information for treated tumors in Fig3d, where authors only show Tconv (the legend also does not explain what Tconv are), and in Fig5e, where authors only show CD4+FoxP3- data. It would be interesting to also show changes in subsets of helper CD4 T cells in the treated tumor site (also w/ anti-CTLA4 Ab) since Treg was no different or increased by anti-CTLA4 Ab with NDV-ICOSL (in Fig5e CD4+Treg+)

2) The anti-tumor effect at the separate tumor site (left) was shown to be robust with NDV-ICOSL + anti-CTLA4 Ab at the primary tumor site (right). This finding suggests the importance of blood circulating immune cells from the point of view of the T cell immune surveillance system. Were there any changes in PBMC before and after the treatments and tumor challenges? Also, has it been confirmed that these distant tumors were not infected by NDV that spilled over from the originally injected tumor?

3) In Figure 2d and 2e, it seems that both NDV-wt and NDV-ICOSL replication capacities are minimum (100 to less than 1000 pfu/ml in B16-F10 cells) and secondary infection (after 24 h p.i.) did seem to occur. Does anti-tumor immunity mediated by NDV require virus replication or is a first round of infection enough? If the latter is true, what is advantage of NDV over PolyI:PolyC or other RNA adjuvants for the immunization? At 0 h pi. virus titer was none: probably this needs to show instead the input amount (pfu in each well or ml). Authors need to provide cell numbers for infection when using m.o.i., so reader can see the inputs used in the experiments.

4) Upregulation of ICOS of immune infiltrates in the NDV-injected tumor makes this an attractive target. However, there is no data to show if tumors injected by NDV-ICOS trigger a similar upregulation. ICOS expression of lymphocytes upon NDV-ICOS infection should be monitored.

5) Does NDV-ICOSL treatment increase FoxP3+ CD4 Treg (Fig1d and Fig5e)?

6) Legend Figure 2: panels b) and c) are switched.

7) What is NDV-GFP in Figure 2f?

It would be preferable to compare NDV-ICOS with NDV-wt unless there are specific reasons since the rest of Fig2 and other experiments used NDV-wt.

8) What effector markers were used for the CD4+ and CD8+ "effector" T cells to distinguish from naïve CD4 or CD8 T cells? Otherwise, "effector" term seems inappropriate.

9) Most mice with tumor challenge at d4 in the distinct flank site reject tumor challenge and were tumor-free (74% by NDV-ICOSL+aCTLA-4 combination in Fig4d). And, Fig5 analysis used d15-mice tissues. Authors need to provide more information about the methods use to isolate and analyze TILs from distinct tumors. Did distinct tumor develop at day 15? It is hard to see any tumor growth from Figure 4d's data.

10) Legend figure 4. It should be stated if mice ave a larger tumor burden before NDV-injection.

11) Fig5a should also include the data of NDV-ICOSL.

- 12) Y-axis scale is incomplete in Fig.1, 3 and 5 FACS analysis data. Probably log10 scale or proper plots to represent the lower number groups. For example, Fig3d Treg population are close to 0. Is it true? Also, would it be possible to express these data as MFIR (calculating the ratio comparing to proper isotype control)?
- 13) Figure 5 data is hard to see the immune cell populations based on the individual mouse. Does mouse with higher number of CD4+FoxP3- show also higher number of CD4+FoxP3+ or lower in Fig 5e? For figure 5b-g, only tests of significance for CTLA4 treated groups are provided. Was test of significances for isotype treated groups performed?.
- 14) Figure 5j. This figure lacks the appropriate controls to conclude if there us tumor specificity. Please add unprimed and, more importantly, irrelevant lysate pulsed DCs to ensure anti-tumor specificity.
- 15) For the discussion: why does NDV-ICOS have better efficacy in combination with CTLA-4 in larger tumors, compared to original oncolytic NDV virus combined with CTLA4.
- 16) Need to provide SD or SEM for the error bars

Reviewer #2 (Remarks to the Author):

The manuscript submitted to Nature Communications by Zamarin et al, entitled "Intratumoral modulation of the inducible costimulator (ICOS) by recombinant oncolytic virus promotes systemic anti-tumor immunity" describes a series of murine experiments in which the authors prepare an attenuated Newcastle disease virus (NDV) variant for intratumoral injection that expresses a transgene for the ICOS-ligand. The senior author has described a series of murine experiments showing that ICOS is important for the therapeutic action of CTLA-4 antibodies alone or in combination, and human data from several groups have shown that ICOS is highly up-regulated on the surface of CD4 and CD8 T cells after CTLA-4 blockade. The laboratory has also shown that local injection of NDV can induce an inflammatory infiltrate and evidence of both local and distant regression in poorly immunogenic tumors. Thus the experiments in the current paper are sensible, are an extension of work previously done in the senior author's lab and have some clinical relevance since the engineered herpesvirus TVEC has recently been approved for local injection in patients with metastatic melanoma.

The authors raise the appropriate concern that larger tumors injected with the wild type NDV followed by systemic CTLA-4 antibody do not really regress distantly as well as smaller tumors, and there is little evidence of benefit. There are no experiments with the ICOS-L expressing virus to show that larger tumors do benefit, however, which is a surprising lapse. In general, most of the work with the non-injected tumors involved 3-4 day B16 lesions, which are very small, and the injected tumors are of a size such that the volume of the injectate easily diffuses around the entire lesion, unlikely to occur in the real-life human situation. Equally surprising is that use of an ICOS knockout strain of mice did not impact on the effect of the wild type NDV + CTLA-4 antibody treatment. No experiments were done with the NDV-ICOS-L virus, which should have been done, since if there were no impact of the elimination of ICOS on the efficacy of the new NDV, then it would require a significant explanation. The fact that there is no explanation provided for the lack of impact of the ICOS knockout on the efficacy of NDV/CTLA-4 treatment is a significant deficiency that reduces the enthusiasm for this work. One would assume it was needed for the efficacy of the NDV-ICOS-L/CTLA-4 treatment, but those experiments are not shown.

The data in figures 4 and 5 are the meat of the work, and while there are data that show a statistically significant improvement in the outcome for the use of the ICOS-L NDV added to CTLA-4, the magnitude is modest, and the even more modest impact on CT-26 is of concern. The data in figure 5 do not provide a satisfactory mechanistic explanation for the improvement in outcome when the NDV-ICOSL construct is added to CYLA-4 antibody, which again weakens enthusiasm for the work.

Additionally, the placement of the figures and their legends in the middle of the text before the discussion was confusing and should be corrected.

Overall, while there appears to be a modest improvement in outcome when a NDV expressing

ICOS-L is locally injected into B16 tumors in association with systemic CTLA-4 blockade, the confusion over the ICOS $-/-$ experiments, the modest therapeutic impact, the lack of impact in the CT26 experiments and the unclear results of the mechanistic data suggest that this work has a level of impact that renders it unsuitable for publication in Nature Communications.

In detail:

Given the lack of impact on regression in the ICOS $-/-$ mice, it is surprising that there are no experiments with the ICOS-L expressing virus to show that injected tumors either do benefit or do not; if the latter, it would change the way the authors view this therapy and would require significant experiments to explain the mechanism.

The placement of the figures and their legends in the middle of the text before the discussion was confusing and should be corrected.

There is no real explanation provided for the lack of impact in the ICOS $-/-$ mice of the NDV WT/CTLA-4 treatment, which merits some verbiage in the discussion.

It would have been useful to combine the TIL from NDV-ICOS-L treated and control NDV-WT treated tumors with ICOS-L transduced or infected B16 tumor cells and NDV WT infected tumor cells as targets to see if the TIL from the experimental tumors were better able to recognize the ICOS-L expressing targets compared with the control NDV infected targets, and if so, would blocking ICOS-ICOS-L interactions with a blocking antibody eliminate the improved recognition.

It would have been useful to combine the TIL from NDV-ICOS-L treated and control NDV-WT treated tumors with ICOS-L transduced or infected B16 tumor cells and NDV WT infected tumor cells as targets to see if the TIL from the experimental tumors were better able to recognize the ICOS-L expressing targets compared with the control NDV infected targets, and if so, would blocking ICOS-ICOS-L interactions with a blocking antibody eliminate the improved recognition.

Reviewers' comments:

Reviewer #1:

The title: Intratumoral modulation of the inducible costimulator (ICOS) by recombinant oncolytic virus promotes systemic anti-tumor immunity.

D. Zamarin et al. found that oncolytic NDV stimulates a robust increase in mDCs at tumor sites and identified that ICOS is significantly higher in TIL amongst all co-stimulation factors analyzed by nanostring. Based on this finding, the authors introduced a ICOS ligand cDNA in a recombinant NDV genome and treated bilateral tumors bearing mice with NDV-ICOSL: the treatment led to significant tumor rejection at a separate flank site when compared with parental NDV-wt and control groups. The authors identified more effector T cells in the TIL of distinct tumor mass. These findings are very interesting and important in immunotherapy using oncolytic viruses because a current limitation in the clinical setting is that therapeutic OV's can only be injected at some tumor sites and cannot be applied to all metastatic sites.

The authors should clarify the following issues:

1) ICOS is thought to be important for helper CD4 subsets (i.e. T follicular, Th1/2/17) during immunization. The authors also show increased CD4 and CD8 T cells at distant tumor sites when NDV-ICOSL was administered to the right flank tumor. However, they do

not provide this information for treated tumors in Fig3d, where authors only show Tconv (the legend also does not explain what Tconv are), and in Fig5e, where authors only show CD4+FoxP3- data. It would be interesting to also show changes in subsets of helper CD4 T cells in the treated tumor site (also w/ anti-CTLA4 Ab) since Treg was no different or increased by anti-CTLA4 Ab with NDV-ICOSL (in Fig5e CD4+Treg+)

Response:

1. As our data were insufficient to dissect the effect of NDV-ICOSL on the different CD4 subsets, per reviewer's suggestion, we have now performed additional experiments to further characterize the different CD4 subsets in the treated tumor in response to NDV-ICOSL and NDV-WT. Of note, in our manuscript we used the terms "T effector (Teff)" and "conventional T cells (Tconv)" interchangeably to refer to CD4+FOXP3- lymphocytes to distinguish them from the FOXP3+ regulatory T cell subset. For consistency, we have changed the term to "conventional T cells (Tconv)" and defined it in the manuscript.

We have performed the analyses looking at the transcription factors defining the different Th lineages (figure 3f) and have specifically focused on the putative Tfh subset, defined as CD4+FOXP3-CXCR5+PD1+ICOS+ (figure 3g). The gating strategy for the latter is shown in Supplementary Figure 10. In comparison to NDV-WT, there were increases in absolute numbers of several Th subsets, including the lymphocytes of Tfh, and Th17 lineages, and a decrease in Th2 cells, though the latter did not reach statistical significance. Despite these findings, the relative percentages of the Tfh subsets out of all CD4+FoxP3- cells were very similar between the two virus groups (Figure 3g). These data suggest that NDV-ICOSL may non-preferentially stimulate the expansion of several Th subsets, including the lymphocytes of cytotoxic, Tfh, and Th17 lineages, while possibly having a negative effect on the expansion of the Th2 lymphocytes. Since these subsets represent a minority of the tumor-infiltrating CD4+FoxP3- cells, their exact role in the anti-tumor response is unclear. Consistent with the increase in therapeutic efficacy, phenotypic characterization of the tumor-infiltrating CD8 cells from both treated and distant tumors revealed a more robust upregulation of the markers indicative of effector function (i.e. ICOS and Granzyme B) (figure 3h). Interestingly, upregulation of Granzyme B coincided with decreased PD-1 expression in these cells (Fig. 3i-j); which may be suggestive of improved effector function, although further studies would be needed to characterize these subsets. Overall, these findings suggest that intratumoral ICOSL expression results in expansion and increased activation status of tumor infiltrating CD8 cells, an effect that is mediated through direct stimulation of the CD8+ cells by ICOSL, or indirectly through the expansion of the Th lineages noted above.

2) The anti-tumor effect at the separate tumor site (left) was shown to be robust with NDV-ICOSL + anti-CTLA4 Ab at the primary tumor site (right). This finding suggests the importance of blood circulating immune cells from the point of view of the T cell immune surveillance system. Were there any changes in PBMC before and after the treatments and tumor challenges? Also, has it been confirmed that these distant tumors were not infected by NDV that spilled over from the originally injected tumor?

Response:

2. We thank the reviewer for the suggestion. We have performed an experiment analyzing the effect of NDV-ICOSL and CTLA-4 blockade on peripheral lymphocytes by characterizing splenic T cell populations (Figure 6 and Supplementary Figures 9 and 10). Treatment with either virus resulted in modest increase in spleen size, but had no substantial effect on the absolute numbers of the splenic CD8+, CD4+FOXP3-, and CD4+FOXP3+ populations. Both NDV-WT and NDV-ICOSL viruses led to expansion of the splenic Tfh populations, defined by expression of Bcl-6 or by CD4+FOXP3-CXCR5+PD1+ICOS+ lineage. While CTLA-4 blockade augmented the expansion of Tfh cells (supporting prior findings of Sage PT et al., *Immunity* 2014, demonstrating the regulatory role of CTLA-4 in Tfh responses), there were no substantial differences seen between the NDV-WT and NDV-ICOSL viruses. However, similar to the data observed in tumors, NDV-ICOSL treatment resulted in increased percentages of splenic CD8+ cells characterized by high expression of Granzyme B and ICOS (Figure 6).

With regards to the question pertaining to the systemic virus spread, we agree that this is a very important question and would like to refer the reviewers to our previously published work where we demonstrate that no virus could be detected in contralateral tumors in this model system (Zamarin et. al, *STM* 2014).

3) In Figure 2d and 2e, it seems that both NDV-wt and NDV-ICOSL replication capacities are minimum (100 to less than 1000 pfu/ml in B16-F10 cells) and secondary infection (after 24 h p.i.) did seem to occur. Does anti-tumor immunity mediated by NDV require virus replication or is a first round of infection enough? If the latter is true, what is advantage of NDV over PolyI:PolyC or other RNA adjuvants for the immunization ? At 0 h pi. virus titer was none: probably this needs to show instead the input amount (pfu in each well or ml). Authors need to provide cell numbers for infection when using m.o.i., so reader can see the inputs used in the experiments.

Response:

3. We appreciate these very thoughtful comments, which are very relevant for the entire field of oncolytic virus therapy. To answer the first comment, we are including more detailed information about the MOI used for virus replication experiments in the Methods section. For the in vitro virus replication experiments, 5e4 cells were infected at MOI of 0.2, with an input virus 1e3. After 1h attachment, the infection media was aspirated, and the cells were incubated with 2ml of fresh media. The amount of residual input virus in the media would be negligible, though some unattached virus would still remain. With these caveats, we agree with the reviewer that the viral concentration at time 0 is likely higher than 0 pfu/ml, however less than 50 pfu/ml (input), and thus excluded this time point from our graphs in figure 2.

With regard to the second comment, the in vitro data indeed demonstrate low-level multi-cycle replication (figure 2d). In our previous published work, we have shown that a single intratumoral injection of NDV expressing luciferase could maintain luciferase signal for several days, suggesting that several cycles of replication likely also take place in vivo (Zamarin et al., *STM* 2014). We believe that sustained viral replication may result in

improved intratumoral spread and delivery of the danger signals to the innate immune effectors, which may enhance activation of the immune response. The main benefit of a replicative virus, however, is the ability of a recombinant NDV (or other oncolytic virus) to deliver a therapeutic transgene (ICOSL in this case), the expression of which can be sustained over a period of time, which is the main premise of this manuscript. We have not conducted any experiments with inactivated NDV to determine whether multi-cycle replication is required for the observed anti-tumor immunity, as inactivated virus would be incapable of expressing a therapeutic transgene. We have included this in the discussion.

4) Upregulation of ICOS of immune infiltrates in the NDV-injected tumor makes this an attractive target. However, there is no data to show if tumors injected by NDV-ICOS trigger a similar upregulation. ICOS expression of lymphocytes upon NDV-ICOS infection should be monitored.

Response:

4. We thank the reviewer for the suggestion. We have collected these data as part of our experiments monitoring for T cell activation in response to NDV and NDV-ICOSL, but did not include them in the original manuscript. We are now including these data in figure 3h and have discussed them as part of the response to the comment #1 above.

5) Does NDV-ICOSL treatment increase FoxP3+ CD4 Treg (Fig1d and Fig5e)?

Response:

5. The reviewer is correct to point out that NDV-ICOSL treatment appears to slightly increase the absolute number of CD4+FOXP3+ cells, which is more pronounced in the NDV-ICOSL-treated group. This could be a compensatory response to the increase in the effector T cell populations, or a known effect of ICOS signaling on expansion of regulatory T cells. The CD4+FOXP3+ cell percentages, however, were not significantly different between the two viral groups and were lower than the percentages seen in the untreated animals or animals treated with CTLA-4 blockade alone. We have included a discussion on this subject on pages 17-18.

6) Legend Figure 2: panels b) and c) are switched.

Response:

6. The figure legend was corrected.

7) What is NDV-GFP in Figure 2f?

It would be preferable to compare NDV-ICOS with NDV-wt unless there are specific reasons since the rest of Fig2 and other experiments used NDV-wt.

Response:

7. The main purpose of the experiment in figure 2f was to determine the percentage of tumor cells that become positive for ICOSL with NDV-ICOSL expression. As the population of tumor cells derived from tumors is very heterogeneous with a broad range in autofluorescence, the use of NDV-GFP allowed us to more accurately estimate the

percentage of infected tumor cells. This closely matched the percentage of cells that became ICOSL+ in the NDV-ICOSL group.

8) What effector markers were used for the CD4+ and CD8+ "effector" T cells to distinguish from naïve CD4 or CD8 T cells? Otherwise, "effector" term seems inappropriate.

Response:

8. In our study, the term "effector" was primarily used to distinguish CD8 and CD4 cells from the CD4+FoxP3+ regulatory T cells. As discussed above, we used the term "Tcon" and "Teff" interchangeably. To clarify this, we have made modifications throughout the manuscript and no longer use the term "effector".

9) Most mice with tumor challenge at d4 in the distinct flank site reject tumor challenge and were tumor-free (74% by NDV-ICOSL+aCTLA-4 combination in Fig4d). And, Fig5 analysis used d15-mice tissues. Authors need to provide more information about the methods used to isolate and analyze TILs from distinct tumors. Did distinct tumor develop at day 15? It is hard to see any tumor growth from Figure 4d's data.

Response:

9. To achieve adequate tumor growth for TIL isolation, the experiments in figure 5 used a significantly larger number of cells than was used for the survival experiments. This was described in the Materials and Methods, section on "Isolation of Tumor-Infiltrating lymphocytes". We have also included a clarification in the figure legend.

10) Legend figure 4. It should be stated if mice ave a larger tumor burden before NDV-injection.

Response:

10. We have modified the legend according to the reviewer's suggestions.

11) Fig5a should also include the data of NDV-ICOSL.

Response:

11. We used figure 5a as a representative panel for the data shown in figure 5b and 5e. According to the reviewer's suggestion, we are also including the panel for NDV-ICOSL.

12) Y-axis scale is incomplete in Fig.1, 3 and 5 FACS analysis data. Probably log10 scale or proper plots to represent the lower number groups. For example, Fig3d Treg population are close to 0. Is it true? Also, would it be possible to express these data as MFIR (calculating the ratio comparing to proper isotype control)?

Response:

12. We thank the reviewer for the suggestion and have modified the scale in these graphs to either a log scale, or to two-segment axes to include lower number groups. With regard to the second comment, not all experiments used an isotype control. For consistency, we left these data as MFI.

13) Figure 5 data is hard to see the immune cell populations based on the individual mouse. Does mouse with higher number of CD4+FoxP3- show also higher number of CD4+FoxP3+ or lower in Fig 5e? For figure 5b-g, only tests of significance for CTLA4 treated groups are provided. Was test of significances for isotype treated groups performed?.

Response:

13. Indeed, there was a strong correlation between the numbers of CD4+FoxP3+ cells and CD4+FoxP3- cells. We have included this clarification in the manuscript. With regard to statistical comparisons, our main hypothesis was comparing the combination-treated groups. We have also performed tests of significance for isotype-treated groups and have found the statistical significance between the isotype- and anti-CTLA-4- combination treated groups to be even higher, given the higher magnitude of the difference. However, we did not include these data in the manuscript as they were not testing the primary hypothesis (comparing the two combination groups) and made the figures look very busy (for the 6 treatment groups, this would result in a total of 14 comparisons).

14) Figure 5j. This figure lacks the appropriate controls to conclude if there is tumor specificity. Please add unpulsed and, more importantly, irrelevant lysate pulsed DCs to ensure anti-tumor specificity.

Response:

14. TRAMP-C2 lysate-pulsed DCs were used as controls in these experiments; the control data were not originally included to simplify the figures. At the request of the reviewers, we are including the relevant control panels for naïve pulsed lymphocytes and TRAMP-C2-pulsed DCs. Because of the size, we moved this figure to the Supplementary Figures section (supplementary figure 8).

15) For the discussion: why does NDV-ICOS have better efficacy in combination with CTLA-4 in larger tumors, compared to original oncolytic NDV virus combined with CTLA4.

Response:

15. We have included this in the discussion on page 17. The findings of this study demonstrate that therapy with NDV-ICOSL and CTLA-4 blockade leads to an expansion and activation of CD8 cells in virus-injected tumors, spleen, and distant tumors. This effect could be mediated by sustained ICOS signaling in the post-priming phase through provision of the ICOSL directly to the CD8 lymphocytes or indirectly, through increase in the intratumoral Th sublineages such as Tfh and Th17 cells. These results mirror previous findings from our group, demonstrating that a cellular tumor vaccine expressing ICOSL (IVAX) was able to significantly increase the efficacy of systemic CTLA-4 blockade against abscopal tumors through enhancement of CD8 lymphocyte function (Fan et. al, JEM 2014).

16) Need to provide SD or SEM for the error bars

Response:

16. This information was included

Reviewer #2 (Remarks to the Author):

The manuscript submitted to Nature Communications by Zamarin et al, entitled "Intratumoral modulation of the inducible costimulator (ICOS) by recombinant oncolytic virus promotes systemic anti-tumor immunity" describes a series of murine experiments in which the authors prepare an attenuated Newcastle disease virus (NDV) variant for intratumoral injection that expresses a transgene for the ICOS-ligand. The senior author has described a series of murine experiments showing that ICOS is important for the therapeutic action of CTLA-4 antibodies alone or in combination, and human data from several groups have shown that ICOS is highly up-regulated on the surface of CD4 and CD8 T cells after CTLA-4 blockade. The laboratory has also shown that local injection of NDV can induce an inflammatory infiltrate and evidence of both local and distant regression in poorly immunogenic tumors. Thus the experiments in the current paper are sensible, are an extension of work previously done in the senior author's lab and have some clinical relevance since the engineered herpesvirus TVEC has recently been approved for local injection in patients with metastatic melanoma. The authors raise the appropriate concern that larger tumors injected with the wild type NDV followed by systemic CTLA-4 antibody do not really regress distantly as well as smaller tumors, and there is little evidence of benefit.

1) There are no experiments with the ICOS-L expressing virus to show that larger tumors do benefit, however, which is a surprising lapse.

Response:

17. In our study, all of the experiments using ICOSL-expressing virus were performed with a larger tumor challenge, the same that was demonstrated in supplementary figure 1. We have included this as a clarification in the manuscript on page 11.

2) In general, most of the work with the non-injected tumors involved 3-4 day B16 lesions, which are very small, and the injected tumors are of a size such that the volume of the injectate easily diffuses around the entire lesion, unlikely to occur in the real-life human situation.

Response:

18. We appreciate the reviewer's concern and agree that this is a problem that affects most of the experiments using the B16 melanoma model, not just by our group, but by all others. In this aggressive tumor model, the kinetics of tumor growth outpaces the kinetics of effective immune response, necessitating earlier treatment. While the model of using small tumors is certainly contrived, it still allows us to compare combinations to the "standard of care" single-agent anti-CTLA-4, which is completely ineffective in this model even against even smaller tumors.

With regard to intratumoral injection, we also agree with the reviewer, but would also like to direct the attention to figure 2f, demonstrating that only a small percentage of tumor cells is infected with virus when tumors of this size are injected, regardless of diffusion around the entire lesion. Furthermore, we have also performed these experiments in the CT26 tumor model, where the starting size of the tumor is ~7- 8mm in diameter. While we agree that even a 7mm tumor still does not approach the larger lesions that would be used in patients for intratumoral injection, mouse transplantable tumor models unfortunately don't allow for significantly larger tumors at the start of treatment, as the tumors continue to grow for several days after therapy initiation before regressing. While these experiments (any mouse model experiment) do not mimic directly the clinical setting they provide proof of principle for the concepts at hand and inform potential future clinical work when interpreted carefully.

3) Equally surprising is that use of an ICOS knockout strain of mice did not impact on the effect of the wild type NDV + CTLA-4 antibody treatment. No experiments were done with the NDV-ICOS-L virus, which should have been done, since if there were no impact of the elimination of ICOS on the efficacy of the new NDV, then it would require a significant explanation. The fact that there is no explanation provided for the lack of impact of the ICOS knockout on the efficacy of NDV/CTLA-4 treatment is a significant deficiency that reduces the enthusiasm for this work. One would assume it was needed for the efficacy of the NDV-ICOS-L/CTLA-4 treatment, but those experiments are not shown.

Response:

19. We agree with the reviewer that the finding that ICOS deficiency does not impair the efficacy of NDV + anti-CTLA-4 antibody treatment is quite surprising, especially since our group and others have demonstrated that ICOS is required for the efficacy of single-agent anti-CTLA-4 therapy. We suspect that other co-stimulatory pathways activated by the combination therapy may be compensating for the ICOS deficiency and have included this in the discussion. With regards to the second comment, NDV-ICOSL in combination with CTLA-4 blockade was tested in ICOS-deficient mice, but we chose not to include these data with the initial submission. At the request of the reviewer, these data are now shown in Supplementary Figure 7. As can be seen from the figure, therapeutic efficacy of NDV-ICOSL in ICOS-deficient mice was virtually identical to that of NDV-WT, suggesting that therapeutic enhancement seen with NDV-ICOSL is indeed dependent on ICOS signaling.

4) The data in figures 4 and 5 are the meat of the work, and while there are data that show a statistically significant improvement in the outcome for the use of the ICOS-L NDV added to CTLA-4, the magnitude is modest, and the even more modest impact on CT-26 is of concern. The data in figure 5 do not provide a satisfactory mechanistic explanation for the improvement in outcome when the NDV-ICOSL construct is added to CTLA-4 antibody, which again weakens enthusiasm for the work.

Response:

20. We agree that the magnitude of the observed benefit is best in the B16 model, is modest in the CT26 model, and is absent in the MB49 model. However, we have specifically opted

to include the data from the several tumor models to demonstrate that the efficacy of NDV-ICOSL seems to relate to the infectivity of the virus and ICOSL expression in these particular tumor cell lines (Figure 2c). The problem of poor infectivity in syngeneic mouse cell lines is not inherent to NDV and has been described for many other viruses.

In regard to the second comment (and requests by reviewer 1), we have performed additional experiments with a goal of characterizing the changes in the intratumoral and systemic lymphocyte populations in response to NDV-ICOSL. As we discussed above, while NDV-ICOSL appears to expand specific T helper subsets in tumors, it is unclear whether this expansion contributes to the overall anti-tumor immune response. Throughout the experiments, we have consistently observed the expansion and increase in activation/lytic markers in the intratumoral and splenic CD8 lymphocytes, suggesting that this is likely the end effector population influenced by this therapy. We have expanded on this in the discussion.

Despite these limitations, we still consider these findings to be important for the following reasons: a) B16-F10 is a very aggressive poorly immunogenic tumor model, which we consider to be a “high bar” and very stringent setting for testing of novel immune therapeutic strategies, as CTLA-4 and PD-1 blockade in this model is completely ineffective; b) even infection of a small percentage of tumor cells with NDV-ICOSL (figure 2f) appears to be sufficient for therapeutic enhancement, highlighting the therapeutic relevance and potency of targeting of this pathway; c) the finding that the expression of ICOSL in human cell lines is orders of magnitude higher than that observed in the mouse cell lines further highlights the therapeutic relevance of this strategy in humans, and d) the study provides a proof of principle that identification of specific targets upregulated in the tumor microenvironment in response to oncolytic virus therapy (or other locoregional approaches) could serve as a rationale for locoregional targeting of such pathways, which may lead to enhanced anti-tumor immunity.

Additionally, the placement of the figures and their legends in the middle of the text before the discussion was confusing and should be corrected.

Response:

21. The journal instructions specifically highlight that the figures may be inserted within the text at the appropriate positions or grouped at the end. In our experience, placement of figures in the middle of the text typically facilitates the review; however, at the reviewer’s request, we have now grouped all of the figures at the end.

Overall, while there appears to be a modest improvement in outcome when a NDV expressing ICOS-L is locally injected into B16 tumors in association with systemic CTLA-4 blockade, the confusion over the ICOS^{-/-} experiments, the modest therapeutic impact, the lack of impact in the CT26 experiments and the unclear results of the mechanistic data suggest that this work has a level of impact that renders it unsuitable for publication in Nature Communications.

Response:

22. We hope we were able to address these concerns through inclusion of additional data and clarifications above.

In detail:

Given the lack of impact on regression in the ICOS^{-/-} mice, it is surprising that there are no experiments with the ICOS-L expressing virus to show that injected tumors either do benefit or do not; if the latter, it would change the way the authors view this therapy and would require significant experiments to explain the mechanism.

See response # 19.

The placement of the figures and their legends in the middle of the text before the discussion was confusing and should be corrected.

See response # 21.

There is no real explanation provided for the lack of impact in the ICOS^{-/-} mice of the NDV WT/CTLA-4 treatment, which merits some verbiage in the discussion.

See response # 19.

It would have been useful to combine the TIL from NDV-ICOS-L treated and control NDV-WT treated tumors with ICOS-L transduced or infected B16 tumor cells and NDV WT infected tumor cells as targets to see if the TIL from the experimental tumors were better able to recognize the ICOS-L expressing targets compared with the control NDV infected targets, and if so, would blocking ICOS-ICOS-L interactions with a blocking antibody eliminate the improved recognition.

Response:

23. The experiment suggested by the reviewer is certainly interesting, however it presents numerous technical challenges, and may be difficult to interpret. First, NDV-infected target tumor cells die from virus-mediated apoptosis, thus assessment of TIL-mediated cytotoxicity against these cells can be problematic, though a cell line expressing ICOSL could be used instead. Second, if enhanced recognition of ICOSL-expressing targets is observed, the recognition could be dependent either on the immunostimulatory effects of ICOSL, the direct recognition of ICOSL as an antigen within the context of MHC, or both. Blocking the ICOSL-ICOS interaction with an antibody in this case might not help to distinguish between these two possibilities, as inhibition of ICOS signaling could potentially inhibit the tumor cell recognition in both cases. Furthermore, development of such an assay would likely require a commitment of a large number of animals and time. The previous studies from our group using ICOSL-expressing tumor cell line vaccines demonstrated that such vaccines did not enhance the efficacy of CTLA-4 blockade when another tumor type was implanted on the contralateral flank (Fan et al., JEM 2014), suggesting that the ICOSL signal needs to be provided in *cis* within the context of the appropriate tumor and that the

ICOSL itself is likely not the target antigen.

Reviewers' comments:

Reviewer #1 (Remarks to the Author):

Dr. Zamarin et al. found that oncolytic NDV stimulates a robust increase in mDCs at tumor sites and identified that ICOS is significantly higher in TIL amongst all co-stimulation factors analyzed by nanostring. Based on this finding, the authors introduced a ICOS ligand cDNA in a recombinant NDV genome and treated bilateral tumors bearing mice with NDV-ICOSL: the treatment led to significant tumor rejection at a separate flank site when compared with parental NDV-wt and control groups. The authors identified more effector T cells in the TIL of distant tumor mass. These findings are very interesting and important in immunotherapy using oncolytic virus because a current limitation in clinical setting is that therapeutic OV's can only be injected at some tumor sites and cannot be applied to all metastatic sites.

General remarks:

1) The observed advantage of expressing of ICOSL was only observed in the distant tumor site, while providing no immediate advantage for the injected tumor (used as monotherapy). This remains odd, as suggested involved immune populations were upregulated in distant and injected tumor. Also, any observed advantage with NDV-ICOSL can not be readily explained, although some advantage can be detected in distant tumors.

2) As it stands, an additional control is needed to confirm that ICOS is the mechanism by which the NDV-ICOSL acts (see remark 5).

Specific Remarks:

1) Legend fig 1 line 797: word appears to be missing (T cell or lymphocytes). This is also the case in some other legends, please verify.

2) Figure 1: It is unclear whether a CD3 staining is performed to plot out CD4+ or CD8+ cells. Or is there are reason it was not included. As it stands, CD3+ is only measured in figure 5. There we can find essential information concerning CD3+ T cells, but there as well CD4+ FoxP3- negative cells is represented as Tcon?

Also, in figure 1 is referred to Teff, while sometimes still referred to as CD4FoxP3- (figure 5) and Tconv (figure 6). Please use similar terminology throughout figures/legends/text.

3) Fig2: (F) the right plot seems a bit strange. It is labelled to plot %ICOS+ cells, could this be % ICOSL+ cells?

4) Line 260-261: should refer to suppl fig 6 instead of 5?

5) Supp Fig 7: NDV-WT + anti-CTLA4 control is lacking in the mice bearing larger tumor.

It is proposed that NDV-ICOSL functions through ICOS signaling. In ICOS-/- KO mice it is indeed observed NDV-ICOSL drops down to near NDV-wt levels. The control of NDV-WT + anti-CTLA4 in ICOS-/- KO mice could determine whether the observed drop in survival of NDV-ICOSL in ICOS-/- KO mice indeed is due to additional effect of ICOSL expression. This would add greatly to confirming the mechanism indeed goes through ICOS. The provided comparison of NDV-wt in wt and ICOS-/- KO mice is not that convincing as indeed this is a model were optimal effect is observed, in which ICOS signaling might not be essential. Therefore, the assumption made in the conclusion 371-373 might be faulty. This should be clarified.

Reviewer #2 (Remarks to the Author):

The authors have generally done a nice job of responding to the concerns and issues of the reviewers, although the fact remains that the impact of the treatment with CTLA-4 with the engineered ICOS-L virus is modest. As pointed out, the therapeutically important figures are 4 and 5, and the panels of important are 4e and 4g; 4G shows a nice difference in OS for the mice injected with B16, but 4g suggest a modest effect with CT26, a fairly immunogenic tumor. There appear to be more CD8 infiltrating cells within tumors with the ICOS construct, but actually more T regs. The data in figure 6 are again modestly consistent for real changes in infiltrating T cells and splenic T cells, and overall while this is well done and careful work, the only concern is the impact of those small changes.

Reviewer #1 (Remarks to the Author):

Dr. Zamarin et al. found that oncolytic NDV stimulates a robust increase in mDCs at tumor sites and identified that ICOS is significantly higher in TIL amongst all co-stimulation factors analyzed by nanostring. Based on this finding, the authors introduced a ICOS ligand cDNA in a recombinant NDV genome and treated bilateral tumors bearing mice with NDV-ICOSL: the treatment led to significant tumor rejection at a separate flank site when compared with parental NDV-wt and control groups. The authors identified more effector T cells in the TIL of distant tumor mass. These findings are very interesting and important in immunotherapy using oncolytic virus because a current limitation in clinical setting is that therapeutic OVs can only be injected at some tumor sites and cannot be applied to all metastatic sites.

General remarks:

1) The observed advantage of expressing of ICOSL was only observed in the distant tumor site, while providing no immediate advantage for the injected tumor (used as monotherapy). This remains odd, as suggested involved immune populations were upregulated in distant and injected tumor. Also, any observed advantage with NDV-ICOSL cannot be readily explained, although some advantage can be detected in distant tumors.

Response:

In the treated tumors, the majority of the inflammatory infiltrate is driven by a response to NDV infection, rather than expression of transgene. While the expression of ICOSL augments the number of TILs and their activation, this appears to be insufficient to drive anti-tumor effect at the injected tumor site, as the reviewer points out. While the mechanism behind this is not entirely clear, it is likely affected by the balance between the tumor-specific TILs and virus-specific TILs. At present it is unknown how expression of ICOSL within the tumor affects this balance. Prior studies with VSV expressing CD40L demonstrated that CD40L can actually decrease the efficacy of VSV by shifting the immune response to predominantly being anti-viral rather than anti-tumor (Galivo et al, Hum Gen Ther 2010). Of note, this is an area of active investigation in the laboratory. We included this in the discussion.

Nevertheless, this by itself is an important finding, suggesting that studies of immunotherapeutic efficacy of recombinant oncolytic viruses solely on the basis of anti-tumor activity in the injected tumors can be misleading. The abscopal effect seen with oncolytic viruses provides a better measure of systemic anti-tumor immunity, and is more likely to be therapeutically relevant.

2) As it stands, an additional control is needed to confirm that ICOS is the mechanism by which the NDV-ICOSL acts (see remark 5).

Response:

See the response to remark 5.

Specific Remarks:

1) Legend fig 1 line 797: word appears to be missing (T cell or lymphocytes). This is also the case in some other legends, please verify.

Response:

Thank you for pointing this out. All of the legends were updated.

2) Figure 1: It is unclear whether a CD3 staining is performed to plot out CD4+ or CD8+ cells. Or is there a reason it was not included. As it stands, CD3+ is only measured in figure 5. There we can find essential information concerning CD3+ T cells, but there as well CD4+ FoxP3-negative cells is represented as Tcon? Also, in figure 1 is referred to Teff, while sometimes still referred to as CD4FoxP3- (figure 5) and Tconv (figure 6). Please use similar terminology throughout figures/legends/text.

Response:

We apologize for the confusion, which likely stems from our inconsistent use of terminology throughout the manuscript. CD3 staining was performed in all experiments. We did not include the quantification of CD3+ cells in figure 1 as it mirrored the increase in CD4+ and CD8+ cells and we did not feel that it was strongly contributory to the data. The plots in Figures 1b and 1c are gated on total CD45+ cells (not CD3) to emphasize the increase in the percentages of tumor-infiltrating CD4+ and CD8+ lymphocytes out of total leukocyte population. The quantifications of lymphocytes in figure 1 (and all figures) are based on CD45+CD3+CD11b- gate. The gating strategy was clarified in the Methods section.

In figure 1, “Teff” refers to CD4+FoxP3- population. Throughout the paper, we refer to the same population as “Tcon”. We now changed the terminology to be consistent throughout the manuscript and the figures.

3) Fig2: (F) the right plot seems a bit strange. It is labelled to plot %ICOS+ cells, could this be %ICOSL+ cells?

Response:

Correct, it should be ICOSL+ cells. We have changed it.

4) Line 260-261: should refer to suppl fig 6 instead of 5?

Response:

Yes, it refers to Supplementary Figure 6. We have updated this.

5) Supp Fig 7: NDV-WT + anti-CTLA4 control is lacking in the mice bearing larger tumor. It is proposed that NDV-ICOSL functions through ICOS signaling. In ICOS-/- KO mice it is indeed observed NDV-ICOSL drops down to near NDV-wt levels. The control of NDV-WT + anti-CTLA4 in ICOS-/- KO mice could determine whether the observed drop in survival of NDV-ICOSL in ICOS-/-KO mice indeed is due to additional effect of ICOSL expression. This would add greatly to confirming the mechanism indeed goes through ICOS. The provided comparison of NDV-wt in wt and ICOS-/- KO mice is not that convincing as indeed this is a model where optimal effect is observed, in which ICOS signaling might not be essential. Therefore, the assumption made in the conclusion 371-373 might be faulty. This should be clarified.

Response:

We agree with the reviewer that inclusion of an additional control (NDV-WT + anti-CTLA-4 in ICOS-KO mice) in the experiment in Supplementary Figure 7c-d would have been useful. Unfortunately, due to poor breeding of ICOS-KO mice, we had significant difficulties generating adequate numbers of mice to perform the experiments described in 7c-d. We feel justified with omitting the NDV-WT+anti-CTLA-4 arm in ICOS-KO mice for the following reasons:

- 1) There was equivalence of NDV+anti-CTLA-4 therapy in WT and ICOS-KO mice with smaller tumor challenge (Supplementary Figure 7a-b). We don't fully agree with the statement that this is a model with optimal effect in which ICOS signaling might not be essential. Since the tumor clearance in this model is not 100%, the model is not fully optimal and we would expect to see at least minor differences if the ICOS pathway was essential.
- 2) In a separate experiment of 5 mice per group, we compared NDV-WT+anti-CTLA-4 in WT and ICOS-KO mice using a larger tumor challenge and found the therapeutic effect to still be equivalent between the groups (albeit at lower efficacy than that seen in Supplementary Figure 7a-b). We did not include these data with our submission due to the small size of the experiment and since we felt that the data are redundant with those presented in Supplementary Figure 7a-b.
- 3) NDV-ICOSL+anti-CTLA-4 combination in ICOS-KO mice appears to be equivalent to NDV-WT+anti-CTLA-4 in WT mice. This argues that lack of ICOS negates the therapeutic effect of ICOSL.
- 4) Previous studies from our group using ICOSL cellular vaccine (IVAX) demonstrated that therapeutic ICOSL solely acted through the ICOS pathway (Fan et al., JEM 2014). These results mirror the results of our current study.

We must, however, agree that we cannot fully exclude that ICOSL could still be acting through additional unidentified pathways, which may be more apparent in a different tumor model. For example, a study by Yao S. et al., has demonstrated that in addition to ICOS, ICOSL has CD28 and CTLA-4 as potential partners (Yao S., et al., Immunity 2011), although this was only observed with human and not mouse molecules. This, however, does not exclude a possibility that mouse ICOSL could have additional interaction partners. We included this as a possibility in discussion.

Reviewer #2 (Remarks to the Author):

The authors have generally done a nice job of responding to the concerns and issues of the reviewers, although the fact remains that the impact of the treatment with CTLA-4 with the engineered ICOS-L virus is modest. As pointed out, the therapeutically important figures are 4 and 5, and the panels of important are 4e and 4g; 4G shows a nice difference in OS for the mice injected with B16, but 4g suggest a modest effect with CT26, a fairly immunogenic tumor. There appear to be more CD8 infiltrating cells within tumors with the ICOS construct, but actually more T regs. The data in figure 6 are again modestly consistent for real changes in infiltrating T cells and splenic T cells, and overall while this is well done and careful work, the only concern is the impact of those small changes.

Response:

We agree that the enhancement is rather modest in the highly-immunogenic CT26 model, though, again, we attribute this to a poor infectivity of the cell line with NDV-ICOSL, resulting in low expression of ICOSL, thus minimizing its contribution to therapeutic effect. We would again like to highlight that the effect is rather strong in the poorly immunogenic B16-F10 model, which is more susceptible to NDV and is a more stringent model for testing of immune therapeutic strategies. Given the good infectivity of NDV in human cell lines, we feel that the B16-F10 model better approximates the therapeutic potential of this strategy in humans.

REVIEWERS' COMMENTS:

Reviewer #1 (Remarks to the Author):

The authors have answered all my questions